# Co-Learning Empirical Games & World Models

## Abstract

Game-based decision-making involves reasoning over both world dynamics and strategic interactions among the agents. Typically, empirical models capturing these respective aspects are learned and used separately. We investigate the potential gain from co-learning these elements: a world model for dynamics and an empirical game for strategic interactions. Empirical games drive world models toward a broader consideration of possible game dynamics induced by a diversity of strategy profiles. Conversely, world models guide empirical games to efficiently discover new strategies through planning. We demonstrate these benefits first independently, then in combination as a new algorithm, Dyna-PSRO, that co-learns an empirical game and a world model. When compared to PSRO—a baseline empirical-game building algorithm, Dyna-PSRO is found to compute lower regret solutions on partially observable general-sum games. In our experiments, Dyna-PSRO also requires substantially fewer experiences than PSRO, a key algorithmic advantage for settings where collecting player-game interaction data is a cost-limiting factor.

## 1 Introduction

Even seemingly simple games can actually embody a level of complexity rendering them intractable to direct reasoning. This complexity stems from the interplay of two sources: dynamics of the game environment, and strategic interactions among the game's players. As an alternative to direct reasoning, models have been developed to facilitate reasoning over these distinct aspects of the game. ***Empirical games*** capture strategic interactions in the form of payoff estimates for joint policies (Wellman, 2006). ***World models*** represent a game's transition dynamics and reward signal directly (Sutton & Barto, 2018; Ha & Schmidhuber, 2018b). Whereas each of these forms of model have been found useful for game reasoning, typical use in prior work has focused on one or the other, learned and employed in isolation from its natural counterpart.

Co-learning both models presents an opportunity to leverage their complementary strengths as a means to improve each other. World models predict successor states and rewards given a game's current state and action(s). However, their performance depends on coverage of their training data, which is limited by the range of strategies considered during learning. Empirical games can inform training of world models by suggesting a diverse set of salient strategies, based on game-theoretic reasoning (Wellman, 2006). These strategies can expose the world model to a broader range of relevant dynamics. Moreover, as empirical games are estimated through simulation of strategy profiles, this same simulation data can be reused as training data for the world model.

Strategic diversity through empirical games, however, comes at a cost. In the popular framework of Policy-Space Response Oracles (PSRO) (Lanctot et al., 2017), empirical normal-form game models are built iteratively, at each step expanding a restricted strategy set by computing best-response policies to the current game's solution. As computing an exact best-response is generally intractable, PSRO uses Deep Reinforcement Learning (DRL) to compute approximate response policies. However, each application of DRL can be considerably resource-intensive, necessitating the generation of a vast amount of gameplays for learning. Whether gameplays, or experiences, are generated via simulation (Obando-Ceron & Castro, 2021) or from real-world interactions (Hester & Stone, 2012), their collection poses a major limiting factor in DRL and by extension PSRO. World models can reduce this cost by transferring previously learned game dynamics across response computations.

We investigate the mutual benefits of co-learning a world model and an empirical game by first verifying the potential contributions of each component independently. We then show how to realize

the combined effects in a new algorithm, *Dyna-PSRO*, that co-learns a world model and an empirical game (illustrated in Figure 1). Dyna-PSRO extends PSRO to learn a world model concurrently with empirical game expansion, and applies this world model to reduce the computational cost of computing new policies.

This is implemented by a Dyna-based reinforcement learner (Sutton, 1990; 1991) that integrates planning, acting, and learning in parallel. Dyna-PSRO is evaluated against PSRO on a collection of partially observable general-sum games. In our experiments, Dyna-PSRO found lower-regret solutions while requiring substantially fewer cumulative experiences.

The main points of novelty of this paper are as follows: (1) empirically demonstrate that world models benefit from the strategic diversity induced by an empirical game; (2) empirically demonstrate that a world model can be effectively transferred and used in planning with new other-players. The major contribution of this work is a new algorithm, Dyna-PSRO, that co-learns an empirical game and world model finding a stronger solution at less cost than the baseline, PSRO.

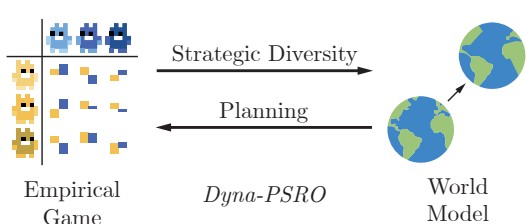

Figure 1: Dyna-PSRO co-learns a world model and empirical game. Empirical games offer world models strategically diverse game dynamics. World models offer empirical games more efficient strategy discovery through planning.

## 2 RELATED WORK

**Empirical Game Theoretic Analysis (EGTA).** The core idea of EGTA (Wellman, 2006) is to reason over approximate game models (*empirical games*) estimated by simulation over a restricted strategy set. This basic approach was first demonstrated by Walsh et al. (2002), in a study of pricing and bidding games. Phelps et al. (2006) introduced the idea of extending a strategy set automatically through optimization, employing genetic search over a policy space. Schvartzman & Wellman (2009a) proposed using RL to derive new strategies that are approximate best responses (BRs) to the current empirical game's Nash equilibrium. The general question of which strategies to add to an empirical game has been termed the *strategy exploration problem* (Jordan et al., 2010). PSRO (Lanctot et al., 2017) generalized the target for BR beyond NE, and introduced DRL for BR computation in empirical games. Many further variants and extensions of EGTA have been proposed, for example those using structured game representations such as extensive-form (McAleer et al., 2021; Konicki et al., 2022). Some prior work has considered transfer learning across BR computations in EGTA, specifically by reusing elements of policies and value functions (Smith et al., 2023a;b; 2021).

**Model-Based Reinforcement Learning (MBRL).** *Model-Based* RL algorithms construct or use a model of the environment (henceforth, *world model*) in the process of learning a policy or value function (Sutton & Barto, 2018). World models may either predict successor observations directly (e.g., at pixel level (Wahlström et al., 2015; Watter et al., 2015)), or in a learned latent space (Ha & Schmidhuber, 2018a; Gelada et al., 2019). World models can be either used for *background planning* by rolling out model-predicted trajectories to train a policy, or for *decision-time planning* where the world model is used to evaluate the current state by planning into the future. Talvitie (2014) demonstrated that even in small Markov decision processes (Puterman, 1994), model-prediction errors tend to compound—rendering long-term planning at the abstraction of observations ineffective. A follow-up study demonstrated that for imperfect models, short-term planning was no better than repeatedly training on previously collected real experiences; however, medium-term planning offered advantages even with an imperfect model (Holland et al., 2018). Parallel studies hypothesized that these errors are a result of insufficient data for that transition to be learned (Kurutach et al., 2018; Buckman et al., 2018). To remedy the data insufficiency, ensembles of world models were proposed to account for world model uncertainty (Buckman et al., 2018; Kurutach et al., 2018; Yu et al., 2020), and another line of inquiry used world model uncertainty to guide exploration in state-action space (Ball et al., 2020; Sekar et al., 2020). This study extends this problem into the multiagent

setting, where now other-agents may preclude transitions from occurring. The proposed remedy is to leverage the strategy exploration process of building an empirical game to guide data generation.

**Multiagent Reinforcement Learning (MARL).** Previous research intersecting MARL and MBRL has primarily focused on modeling the opponent, particularly in scenarios where the opponent is fixed and well-defined. Within specific game sub-classes, like cooperative games and two-player zero-sum games, it has been theoretically shown that opponent modeling reduces the sample complexity of RL (Tian et al., 2019; Zhang et al., 2020). Opponent models can either explicitly (Mealing & Shapiro, 2015; Foerster et al., 2018) or implicitly (Bard et al., 2013; Indarjo, 2019) model the behavior of the opponent. Additionally, these models can either construct a single model of opponent behavior, or learn a set of models (Collins, 2007; He et al., 2016). While opponent modeling details are beyond the scope of this study, readers can refer to Albrecht & Stone's survey (Albrecht & Stone, 2018) for a comprehensive review on this subject. Instead, we consider the case where the learner has explicit access to the opponent's policy during training, as is the case in empirical-game building. A natural example is that of Self-Play, where all agents play the same policy; therefore, a world model can be learned used to evaluate the quality of actions with Monte-Carlo Tree Search (Silver et al., 2016; 2017; Tesauro, 1995; Schrittwieser et al., 2020). Li et al. (2023) expands on this by building a population of candidate opponent policies through PSRO to augment the search procedure. Krupnik et al. (2020) demonstrated that a generative world model could be useful in multi-step opponent-action prediction. Sun et al. (2019) examined modeling stateful game dynamics from observations when the agents' policies are stationary. Chockingam et al. (2018) explored learning world models for homogeneous agents with a centralized controller in a cooperative game. World models may also be shared by independent reinforcement learners in cooperative games (Willemsen et al., 2021; Zhang et al., 2022).

## 3 Co-Learning Benefits

We begin by specifying exactly what we mean by world model and empirical game. This requires defining some primitive elements. Let $t \in \mathcal{T}$ denote time in the real game, with $s^t \in \mathcal{S}$ the **information state** and $h^t \in \mathcal{H}$ the **game state** at time $t$. The information state $s^t \equiv (m^{\pi,t}, o^t)$ is composed of the **agent's memory** $m^\pi \in \mathcal{M}^\pi$, or recurrent state, and the current **observation** $o \in \mathcal{O}$. Subscripts denote a player-specific component $s_i$, negative subscripts denote all but the player $s_{-i}$, and boldface denote the joint of all players $\boldsymbol{s}$. The **transition dynamics** $p : \mathcal{H} \times \boldsymbol{\mathcal{A}} \to \Delta(\mathcal{H}) \times \Delta(\mathcal{R})$ define the game state update and reward signal. The agent experiences **transitions**, or **experiences**, $(s^t, a^t, r^{t+1}, s^{t+1})$ of the game; where, sequences of transitions are called **trajectories** $\tau$ and trajectories ending in a terminal game state are **episodes**.

At the start of an episode, all players sample their current **policy** $\pi$ from their **strategy**[1] $\sigma : \Pi \to [0,1]$, where $\Pi$ is the **policy space** and $\Sigma$ is the corresponding **strategy space**. A **utility function** $U : \boldsymbol{\Pi} \to \mathbb{R}^n$ defines the payoffs/returns (i.e., cumulative reward) for each of $n$ players. The tuple $\Gamma \equiv (\boldsymbol{\Pi}, U, n)$ defines a **normal-form game** (NFG) based on these elements. We represent empirical games in normal form. An **empirical normal-form game** (ENFG) $\hat{\Gamma} \equiv (\hat{\boldsymbol{\Pi}}, \hat{U}, n)$ models a game with a **restricted strategy set** $\hat{\boldsymbol{\Pi}}$ and an estimated payoff function $\hat{U}$. An empirical game is typically built by alternating between game reasoning and strategy exploration. During the game reasoning phase, the empirical game is solved based on a solution concept predefined by the modeler. The strategy exploration step uses this solution to generate new policies to add to the empirical game. One common heuristic is to generate new policies that best-respond to the current solution (McMahan et al., 2003; Schvartzman & Wellman, 2009b). As exact best-responses typically cannot be computed, RL or DRL are employed to derive approximate best-responses (Lanctot et al., 2017).

An **agent world model** $w$ represents dynamics in terms of information available to the agent. Specifically, $w$ maps observations and actions to observations and rewards for all agents, $w : \mathcal{O} \times \boldsymbol{\mathcal{A}} \times \mathcal{M}^w \to \mathcal{O} \times \mathcal{R}$, where $m^w \in \mathcal{M}^w$ is the **world model's memory**, or recurrent state. For simplicity, in this work, we assume the agents learn and use a shared deterministic world model, irrespective of stochasticity that may be present in the true game. Implementation details are provided in Appendix C.2.

Until now, we have implicitly assumed the need for distinct models. However, if a single model could serve both functions, co-learning two separate models would not be needed. Empirical games, in

---

[1]This is equivalent to the meta-strategy defined in PSRO (Lanctot et al., 2017).

general, cannot replace a world model as they entirely abstract away any concept of game dynamics. Conversely, world models have the potential to substitute for the payoff estimations in empirical games by estimating payoffs as rollouts with the world model. We explore this possibility in an auxiliary experiment included in Appendix E.4, but our findings indicate that this substitution is impractical. Due to compounding of model-prediction errors, the payoff estimates and entailed game solutions were quite inaccurate.

Having defined the models and established the need for their separate instantiations, we can proceed to evaluate the claims of beneficial co-learning. Our first experiment shows that the strategic diversity embodied in an empirical game yields diverse game dynamics, resulting in the training of a more performant world model. The second set of experiments demonstrates that a world model can help reduce the computational cost of policy construction in an empirical game.

## 3.1 STRATEGIC DIVERSITY

A world model is trained to predict successor observations and rewards, from the current observations and actions, using a supervised learning signal. Ideally, the training data would cover all possible transitions. This is not feasible, so instead draws are conventionally taken from a dataset generated from play of a ***behavioral [joint] strategy***. Performance of the world model is then measured against a ***target [joint] strategy***. Differences between the behavioral and target strategies present challenges in learning an effective world model.

We call the probability of drawing a state-action pair under some joint strategy its ***reach probability***. From this, we define a joint strategy's ***strategic diversity*** as the distribution induced from reach probabilities across the full state-action space. These terms allow us to observe two challenges for learning world models. First, the diversity of the behavioral joint strategy ought to *cover* the target joint strategy's diversity. Otherwise, transitions will be absent from the training data. It is possible to supplement coverage of the absent transitions if they can be generalized from covered data; however, this cannot be generally guaranteed. Second, the *closer* the diversities are, the more accurate the learning objective will be. An extended description these challenges is provided in Appendix C.3.

If the target joint strategy were known, we could readily construct the ideal training data for the world model. However the target is generally not known at the outset; indeed determining this target is the ultimate purpose of empirical game reasoning. The evolving empirical game essentially reflects a search for the target. Serendipitously, construction of this empirical game entails generation of data that captures elements of likely targets. This data can be reused for world model training without incurring any additional data collection cost.

**Game.** We evaluate the claims of independent co-learning benefits within the context of a *commons game* called "Harvest". In Harvest, players move around an orchard picking apples. The challenging commons element is that apple regrowth rate is proportional to nearby apples, so that socially optimum behavior would entail managed harvesting. Self-interested agents capture only part of the benefit of optimal growth, thus non-cooperative equilibria tend to exhibit collective over-harvesting. The game has established roots in human-behavioral studies (Janssen et al., 2010) and in agent-based modeling of emergent behavior (Pérolat et al., 2017; Leibo et al., 2017; 2021). For our initial experiments, we use a symmetric two-player version of the game, where in-game entities are represented categorically (HumanCompatibleAI, 2019). Each player has a $10 \times 10$ viewbox within their field of vision. The possible actions include moving in the four cardinal directions, rotating either way, tagging, or remaining idle. A successful tag temporarily removes the other player from the game, but can only be done to other nearby players. Players receive a reward of $1$ for each apple picked. More detailed information and visualizations are available in Appendix D.1.

**Experiment.** To test the effects of strategic diversity, we train a suite of world models that differ in the diversity of their training data. The datasets are constructed from the play of three policies: a random baseline policy, and two PSRO-generated policies. The PSRO policies were arbitrarily sampled from an approximate solution produced by a run of PSRO. We sampled an additional policy from PSRO for evaluating the generalization capacity of the world models. These policies are then subsampled and used to train seven world models. The world models are referred to by icons ⊞ that depict the symmetric strategy profiles used to train them in the normal-form. Strategy profiles included in the training data of the world models are shaded black. For instance, the first

(random) policy ▦, or the first and third policies ▦. Each world model's dataset contains 1 million total transitions, collected uniformly from each distinct strategy profile (symmetric profiles are not re-sampled). The world models are then evaluated on accuracy and recall for their predictions of both observation and reward for both players. The world models are optimized with a weighted-average cross-entropy objective. Additional details are in Appendix C.2.

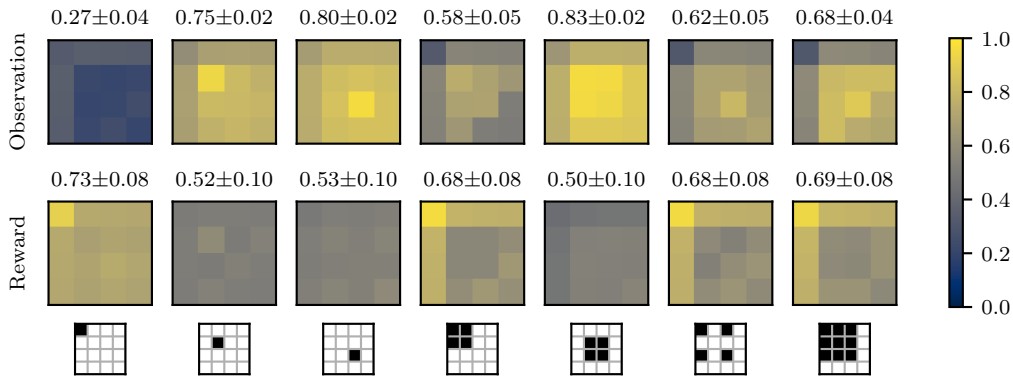

Figure 2: World model accuracy across strategy profiles. Each heatmap portrays a world model's accuracy over 16 strategy profiles. The meta x-axis corresponds to the profiles used to train the world model (as black cells). Above each heatmap is the model's average accuracy.

**Results.** Figure 2 presents each world model's per-profile accuracy, as well as its average over all profiles. Inclusion of the random policy corresponds to decreases in observation prediction accuracy: ▦ $0.75 \pm 0.02 \rightarrow$ ▦ $0.58 \pm 0.05$, ▦ $0.80 \pm 0.02 \rightarrow$ ▦ $0.62 \pm 0.05$, and ▦ $0.83 \pm 0.02 \rightarrow$ ▦ $0.68 \pm 0.04$. Figure 14 (Appendix E.1) contains the world model's per-profile recall. Inclusion of the random policy corresponds to increases in reward 1 recall: ▦ $0.25 \pm 0.07 \rightarrow$ ▦ $0.37 \pm 0.11$, ▦ $0.25 \pm 0.07 \rightarrow$ ▦ $0.36 \pm 0.11$, and ▦ $0.26 \pm 0.07 \rightarrow$ ▦ $0.37 \pm 0.11$.

**Discussion.** The PSRO policies offer the most strategically salient view of the game's dynamics. Consequently, the world model ▦ trained with these policies yields the highest observation accuracy. However, this model performs poorly on reward accuracy, scoring only $0.50 \pm 0.10$. In comparison, the model trained on the random policy ▦ scores $0.73 \pm 0.08$. This seemingly counterintuitive result can be attributed to a significant class imbalance in rewards. ▦ predicts only the most common class, no reward, which gives the illusion of higher performance. In contrast, the remaining world models attempt to predict rewarding states, reducing their overall accuracy. Therefore, we should compare the world models based on their ability to recall rewards. When we examine ▦ again, we find that it also struggles to recall rewards, scoring only $0.26 \pm 0.07$. However, when the random policy is included in the training data (▦), the recall improves to $0.37 \pm 0.11$. This improvement is also due to the same class imbalance. The PSRO policies are highly competitive, tending to over-harvest. This limits the proportion of rewarding experiences. Including the random policy enhances the diversity of rewards in this instance, as its coplayer can demonstrate successful harvesting. Given the importance of accurately predicting both observations and rewards for effective planning, ▦ appears to be the most promising option. However, the strong performance of ▦ suggests future work on algorithms that can benefit solely from observation predictions. Overall, these results provide some evidence supporting the claim that strategic diversity enhances the training of world models.

## 3.2 RESPONSE CALCULATIONS

Empirical games are built by iteratively calculating and incorporating responses to the current solution. However, direct computation of these responses is often infeasible, so RL or DRL is used to approximate the response. This process of approximating a single response policy using RL is computationally intensive, posing a significant constraint in empirical game modeling when executed

repeatedly. World models present an opportunity to address this issue. A world model can serve as a medium for transferring previously learned knowledge about the game's dynamics. Therefore, the dynamics need not be relearned, reducing the computational cost associated with response calculation.

Exercising a world model for transfer is achieved through a process called ***planning***. Planning is any procedure that takes a world model and produces or improves a policy. In the context of games, planning can optionally take into account the existence of coplayers. This consideration can reduce experiential variance caused by unobserved confounders (i.e., the coplayers). However, coplayer modeling errors may introduce further errors in the planning procedure (He et al., 2016).

Planning alongside empirical-game construction allows us to side-step this issue as we have direct access to the policies of all players during training. This allows us to circumvent the challenge of building accurate agent models. Instead, the policies of coplayers can be directly queried and used alongside a world model, leading to more accurate planning. In this section, we empirically demonstrate the effectiveness of two methods that decrease the cost of response calculation by integrating planning with a world model and other agent policies.

### 3.2.1 BACKGROUND PLANNING

The first type of planning that is investigated is ***background planning***, popularized by the Dyna architecture (Sutton, 1990). In background planning, agents interact with the world model to produce ***planned experiences***[2]. The planned experiences are then used by a model-free reinforcement learning algorithm as if they were ***real experiences*** (experiences generated from the real game). Background planning enables learners to generate experiences of states they are not currently in.

**Experiment.** To assess whether planned experiences are effective for training a policy in the actual game, we compute two response policies. The first response policy, our baseline, learns exclusively from real experiences. The second response policy, referred to as the planner, is trained using a two-step procedure. Initially, the planner is exclusively trained on planned experiences. After 10 000 updates, it then transitions to learning solely from real experiences. Policies are trained using IMPALA (Espeholt et al., 2018), with further details in Appendix C.1. The planner employs the ▉ world model from Section 3.1, and the opponent plays the previously held-out policy. In this and subsequent experiments, the cost of methods is measured by the number of experiences they require with the actual game. Throughout the remainder of this work, each experience represents a trajectory of 20 transitions, facilitating the training of recurrent policies.

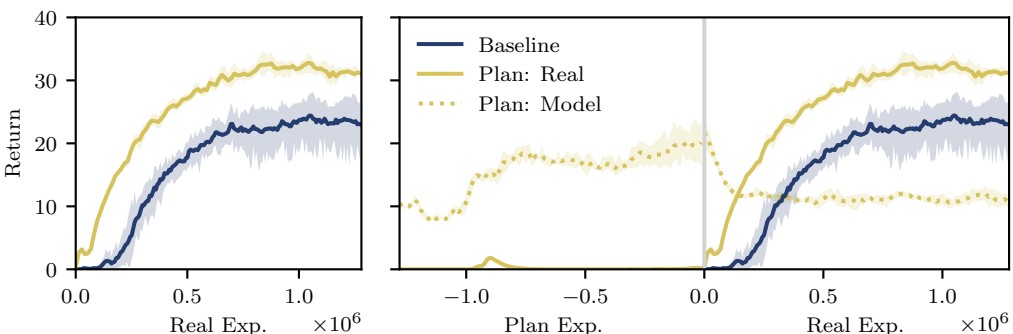

Figure 3: Effects of background planning on response learning. Left: Return curves measured by the number of real experiences used. Right: Return curves measured by usage of both real and planned experiences. The planner's return is measured against the real game (Plan: Real) and the world model (Plan: Model). (5 seeds, with 95 % bootstrapped CI).

**Results.** Figure 3 presents the results of the background planning experiment. The methods are compared based on their final return, utilizing an equivalent amount of real experiences. The baseline yields a return of $23.00 \pm 4.01$, whereas the planner yields a return of $31.17 \pm 0.25$.

---

[2]Other names include "imaginary", "simulated", or "hallucinated" experiences.

**Discussion.** In this experiment, the planner converges to a stronger policy, and makes earlier gains in performance than the baseline. Despite this, there is a significant gap in the planner's learning curves, which are reported with respect to both the world model and real game. This gap arises due to accumulated model-prediction errors, causing the trajectories to deviate from the true state space. Nevertheless, the planner effectively learns to interact with the world model during planning, and this behavior shows positive transfer into the real game, as evidenced by the planner's rapid learning. The exact magnitude of benefit will vary across coplayers' policies, games, and world models. In Figure 16 (Appendix E.2), we repeat the same experiment with the poorly performing ▦ world model, and observe a marginal benefit ($26.05 \pm 1.32$). The key take-away is that background planning tends to lead towards learning benefits, and not generally hamper learning.

### 3.2.2 DECISION-TIME PLANNING

The second main way that a world model is used is to inform action selection at ***decision time [planning] (DT)***. In this case, the agent evaluates the quality of actions by comparing the value of the model's predicted successor state for all candidate actions. Action evaluation can also occur recursively, allowing the agent to consider successor states further into the future. Overall, this process should enable the learner to select better actions earlier in training, thereby reducing the amount of experiences needed to compute a response. A potential flaw with decision-time planning is that the agent's learned value function may not be well-defined on model-predicted successor states (Talvitie, 2014). To remedy this issue, the value function should also be trained on model-predicted states.

**Experiment.** To evaluate the impact the decision-time planning, we perform an experiment similar to the background planning experiment (Section 3.2.1). However, in this experiment, we evaluate the quality of four types of decision-time planners that perform one-step three-action search. The planners differ in the their ablations of background planning types: (1) ***warm-start background planning (BG: W)*** learning from planned experiences before any real experiences, and (2) ***concurrent background planning (BG: C)*** where after BG: W, learning proceeds simultaneously on both planned and real experiences. The intuition behind BG: C is that the agent can complement its learning process by incorporating planned experiences that align with its current behavior, offsetting the reliance on costly real experiences. Extended experimental details are provided in Appendix C.

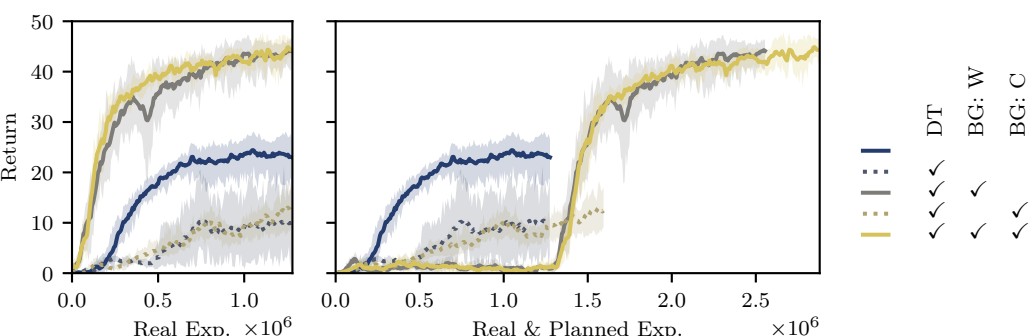

Figure 4: Effects of decision-time planning on response learning. Four planners using decision-time planning (DT) are shown in combinations with warm-start background planning (BG: W) and concurrent background planning (BG: C). (5 seeds, with $95\%$ bootstrapped CI).

**Results.** The results for this experiment are shown in Figure 4. The baseline policy receives a final return of $23.00 \pm 4.01$. The planners that do not include BG: W, perform worse, with final returns of $9.98 \pm 7.60$ (DT) and $12.42 \pm 3.97$ (DT & BG: C). The planners that perform BG: W outperform the baseline, with final returns of $44.11 \pm 2.81$ (DT & BG: W) and $44.31 \pm 2.56$ (DT, BG: W, & BG: C).

**Discussion.** Our results suggest that the addition of BG: W provides sizable benefits: $9.98 \pm 7.60$ (DT) $\rightarrow 44.11 \pm 2.81$ (DT & BG:W) and $12.42 \pm 3.97$ (DT & BG: C) $\rightarrow 44.31 \pm 2.56$ (DT, BG: W, & BG: C). We postulate that this is because it informs the policy's value function on model-predictive

states early into training. This allows that the learner is able to more effectively search earlier into training. BG: C appears to offer minor stability and variance improvements throughout the training procedure; however, it does not have a measurable difference in final performance.

However, we caution against focusing on the magnitude of improvement found within this experiment. As the margin of benefit depends on many factors including the world model accuracy, the opponent policy, and the game. To exemplify, similar to the background planning section, we repeat the same experiment with the poorly performing ▦ world model. The results of this ancillary experiment are in Figure 18 (Appendix E.3). The trend of BG: W providing benefits was reinforced: $6.29 \pm 5.12$ (DT) $\rightarrow 20.98 \pm 9.76$ (DT & BG: W) and $3.64 \pm 0.26$ (DT & BG: C) $\rightarrow 33.07 \pm 7.67$ (DT, BG: W, & BG: C). However, the addition of BG: C now measurably improved performance $20.98 \pm 9.76$ (DT & BG: W) $\rightarrow 33.07 \pm 7.67$ (DT, BG: W, & BG: C). The main outcome of these experiments is the observation that *multi-faceted* planning is unlikely to harm a response calculation, and has a potentially large benefit when applied effectively. In total, the results of this experiment support the claim that world models offer the potential to improve response calculation through DT planning.

# 4 DYNA-PSRO

In this section we introduce Dyna-PSRO, *Dyna*-Policy-Space Response Oracles, an approximate game-solving algorithm that builds on the PSRO (Lanctot et al., 2017) framework. Dyna-PSRO employs co-learning to combine the benefits of world models and empirical games.

Dyna-PSRO is defined by two significant alterations to the original PSRO algorithm. First, it trains a world model in parallel with all the typical PSRO routines (i.e., game reasoning and response calculation). We collect training data for the world model from both the episodes used to estimate the empirical game's payoffs, and the episodes that are generated during response learning and evaluation. This approach ensures that the world model is informed by a diversity of data from a salient set of strategy profiles. By reusing data from empirical game development, training the world model incurs no additional cost for data collection.

The second modification introduced by Dyna-PSRO pertains to the way response policies are learned. Dyna-PSRO adopts a Dyna-based reinforcement learner (Sutton, 1990; 1991; Sutton et al., 2012) that integrates simultaneous planning, learning, and acting. Consequently, the learner concurrently processes experiences generated from decision-time planning, background planning, and direct game interaction. These experiences, regardless of their origin, are then learned from using the IMPALA (Espeholt et al., 2018) update rule. For all accounts of planning, the learner uses the single world model that is trained within Dyna-PSRO. This allows game knowledge accrued from previous response calculations to be transferred and used to reduce the cost of the current and future response calculations. Pseudocode and additional details are provided in Appendix C.4.

**Games.** Dyna-PSRO is evaluated on three games. The first is the harvest commons game used in the experiments described above, denoted "Harvest: Categorical". The other two games come from the MeltingPot (Leibo et al., 2021) evaluation suite and feature image-based observations. "Harvest: RGB" is their version of the same commons harvest game (details in Appendix D.2). "Running With Scissors" is a temporally extended version of rock-paper-scissors (details in Appendix D.3). World model details for each game are in Appendix C.2, likewise, policies in Appendix C.1.

**Experiment.** Dyna-PSRO's performance is measured by the quality of the Nash equilibrium solution it produces when compared against the world-model-free baseline PSRO. The two methods are evaluated on SumRegret (sometimes called *Nash convergence*), which measures the regret across all players $\text{SumRegret}(\boldsymbol{\sigma}, \overline{\boldsymbol{\Pi}}) = \sum_{i \in n} \max_{\pi_i \in \overline{\Pi}_i} \hat{U}_i(\pi_i, \sigma_{-i}) - \hat{U}_i(\sigma_i, \sigma_{-i})$, where $\boldsymbol{\sigma}$ is the method's solution and $\overline{\overline{\Pi}} \subseteq \boldsymbol{\Pi}$ denotes the deviation set. We define deviation sets based on policies generated across methods: $\overline{\overline{\Pi}} \equiv \bigcup_{\text{method}} \hat{\boldsymbol{\Pi}}^{\text{method}}$ (i.e., regret is with respect to the *combined game*—a union of each method's empirical game), for all methods for a particular seed (detailed in Appendix C.5) (Balduzzi et al., 2018). We measure SumRegret for intermediate solutions, and report it as a function of the cumulative number of real experiences employed in the respective methods.

**Results.** Figure 5 presents the results for this experiment. For Harvest: Categorical, Dyna-PSRO found a no regret solution within the combined-game in 3.2e6 experiences. Whereas, PSRO achieves

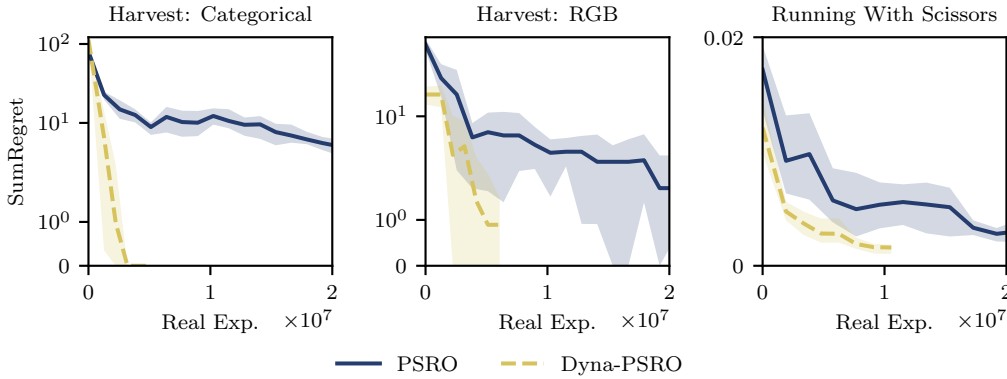

Figure 5: PSRO compared against Dyna-PSRO. (5 seeds, with $95\,\%$ bootstrapped CI).

a solution of at best $5.45 \pm 1.62$ within 2e7 experiences. In Harvest: RGB, Dyna-PSRO reaches a solution with $0.89 \pm 0.74$ regret at 5.12e6 experiences. At the same time, PSRO had found a solution with $6.42 \pm 4.73$ regret, and at the end of its run had $2.50 \pm 2.24$ regret. In the final game, RWS, Dyna-PSRO has $2\mathrm{e}{-3} \pm 5\mathrm{e}{-4}$ regret at 1.06e7 experiences, and at a similar point (9.6e6 experiences), PSRO has $6.68\mathrm{e}{-3} \pm 2.51\mathrm{e}{-3}$. At the end of the run, PSRO achieves a regret $3.50\mathrm{e}{-3} \pm 7.36\mathrm{e}{-4}$.

**Discussion.**    The results indicate that across all games, Dyna-PSRO consistently outperforms PSRO by achieving a superior solution. Furthermore, this improved performance is realized while consuming fewer real-game experiences. For instance, in the case of Harvest: Categorical, the application of the world model for decision-time planning enables the computation of an effective policy after only a few iterations. On the other hand, we observe a trend of accruing marginal gains in other games, suggesting that the benefits are likely attributed to the transfer of knowledge about the game dynamics. In Harvest: Categorical and Running With Scissors, Dyna-PSRO also had lower variance than PSRO.

## 5    CONCLUSION & LIMITATIONS

This study showed the mutual benefit of co-learning a world model and empirical game. First, we demonstrated that empirical games provide strategically diverse training data that could inform a more robust world model. We then showed that world models can reduce the computational cost, measured in experiences, of response calculations through planning. These two benefits were combined and realized in a new algorithm, Dyna-PSRO. In our experiments, Dyna-PSRO computed lower-regret solutions than PSRO on several partially observable general-sum games. Dyna-PSRO also required substantially fewer experiences than PSRO, a key algorithmic advantage for settings where collecting experiences is a cost-limiting factor.

Although our experiments demonstrate benefits for co-learning world models and empirical games, there are several areas for potential improvement. The world models used in this study necessitated observational data from all players for training, and assumed a simultaneous-action game. Future research could consider relaxing these assumptions to accommodate different interaction protocols, a larger number of players, and incomplete data perspectives. Furthermore, our world models functioned directly on agent observations, which made them computationally costly to query. If the generation of experiences is the major limiting factor, as assumed in this study, this approach is acceptable. Nevertheless, reducing computational demands through methods like latent world models presents a promising avenue for future research. Lastly, the evaluation of solution concepts could also be improved. While combined-game regret employs all available estimates in approximating regret, its inherent inaccuracies may lead to misinterpretations of relative performance.

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
