## A    BROADER IMPACT

There are no direct broader impacts from this work. However, this work promotes the adoption of both empirical games and world models. Potential negative impacts may arise due to errors introduced when compressing the true game into the confines of *any* model, which could lead to negative consequences. World model errors within Dyna-PSRO are transferred across response calculations potentially reinforcing biases about the world. If these biases are not rectified, they could negatively influence policies learned from these models. The strategic diversity component of this work aims to mitigate these potential biases, though it represents only the initial step in addressing this concern. When considering empirical games, inaccuracies within them can lead to the suggestion of flawed solutions. The adoption of these inaccurate solutions could have negative repercussions for practitioners or other stakeholders involved in the game. Vigilance and thorough evaluation are required to prevent these potential issues.

## B    COMPUTE

GPUs are used for training world models, and policies within Dyna-PSRO. Two types of GPUs were used throughout this work interchangeably: TITAN X and GTX 1080 Ti. All other computation was completed using CPUs. Each response calculation had additional CPUs corresponding to the number of experience generation arenas described in Appendix C. Experiments were run on internal clusters.

## C    METHODS DETAILS

In this work, the both the policies and world models are implemented in JAX (Bradbury et al., 2018) with Haiku (Hennigan et al., 2020). The software is architected using Launchpad (Yang et al., 2021) with design patterns inspired by ACME (Hoffman et al., 2020). All replay buffers are implemented using Reverb (Cassirer et al., 2021). Gambit (McKelvey et al., 2016) is used as a game solver via linear complementarity (Eaves, 1971).

### C.1    POLICY IMPLEMENTATION & TRAINING

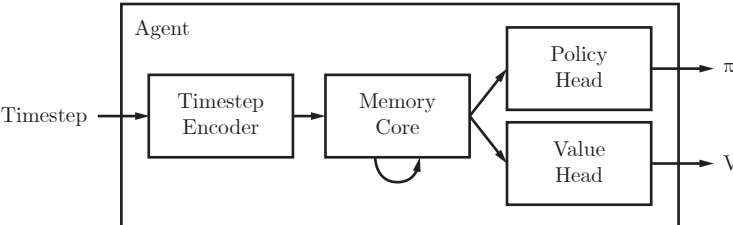

Figure 6: Agent Architecture.

All policies follow the general architecture depicted in Figure 6. This consists of four modules:

- *Timestep Encoder*: Processes all of the current observation's information into a single embedding vector. The timestep includes the new observation and the policy's previous action.
- *Memory Core*: The component of the agent that maintains and update's the agent's memory.
- *Policy Head*: Computes the agent's policy.
- *Value Head*: Computes the agent's state value function.

All of the components are simultaneously trained and their joint parameters are $\theta^\pi \in \Theta^\pi$. The policies are trained using the IMPALA algorithm (Espeholt et al., 2018). For the IMPALA loss, the coefficients for each component loss are:

$$\mathcal{L}_{\text{IMPALA}} = \lambda_\pi \cdot \mathcal{L}_\pi + \lambda_{\text{V}} \cdot \mathcal{L}_{\text{V}} + \lambda_{\text{entropy}} \cdot \mathcal{L}_{\text{entropy}},$$

with a discount factor of 0.99. The training details for each specific response calculation are itemized below.

**Baseline Parameters** The learning rate begins is linearly decayed over $10\,000$ updates. Each update is computed from a mini-batch of $128$ examples that are generated from 8 arenas[3] Policy parameters are synchronized at the beginning of each episode. Each example in the mini-batch is a sequence of 20 transitions. Moreover, sequences are stored in a replay buffer with a period of 19, to ensure that the action played at the end of a sequence is trained. Sequences are stored in a replay buffer with a max capacity of $1\,000\,000$, and are evicted once sampled. Additional hyperparameters are specified in Table 1.

Table 1: Baseline policy hyperparameters per game.

| Hyperparameter | Harvest: Categorical | Harvest: RGB | Running with Scissors |
|---|---|---|---|
| Optimizer | Adam | RMSProp | RMSProp |
| $\lambda_\pi$ | 1.0 | 1.0 | 1.0 |
| $\lambda_V$ | 0.2 | 0.5 | 0.2 |
| $\lambda_{\text{entropy}}$ | 0.04 | 0.01 | 0.003 |
| Learning Rate Start | 6e−6 | 6e−4 | 1e−4 |
| Learning Rate Stop | 6e−9 | 6e−9 | 1e−4 |
| Max Grad Norm | 10.0 | 1.0 | 0.1 |
| Batch Size | 128 | 128 | 128 |

Harvest: Categorical module implementations:

- *Timestep Encoder*: The encoder processes two timestep components: the current observation and the previous action the policy took. First the observation is passed through a two-layer fully connected neural network with hidden sizes of $[256, 256]$. The representation of the observation is then concatenated with the previous action (represented as a one-hot vector), and passed together through a second neural network with sizes $[256, 256]$. All of the layers have ReLU (Fukushima, 1975) activations including the final layers of both networks. The final representation is the output of the timestep encoder.
- *Memory Core*: A single-layer LSTM (Hochreiter & Schmidhuber, 1997) with $256$ units.
- *Policy Head*: A single linear layer of size 8.
- *Value Head*: A single linear layer of size 1.

Harvest: RGB and Running with Scissors module implementations:

- *Timestep Encoder*: The encoder processes two timestep components: the current observation and the previous action the policy took. The observation is first process by a two-layer convolutional neural network with ReLU activations (Fukushima, 1975). The first layer has 16 channels, a kernel with shape $[8, 8]$, and a stride of $[8, 8]$. The second layer has 32 channels, a kernel shape of $[4, 4]$, and a stride of $[1, 1]$. The output of this layer is then flattened and concatenated with a one-hot encoding of the policy's previous action. The resulting embedding is then passed through a two-layer fully connected neural network with hidden sizes of $[128, 128]$, and ReLU activations.
- *Memory Core*: A single-layer LSTM (Hochreiter & Schmidhuber, 1997) with $128$ units.
- *Policy Head*: A single linear layer of size 8.
- *Value Head*: A single linear layer of size 1.

**Planning Parameters** The planners have the same hyperparameters as the baseline method, but with the addition of planning-specific settings. For all planners, an additional 4 arenas are used to generate planned experiences (for background planning). The additional settings for each version of planning are as follows:

---

[3]The term *arena* is used to refer to an experience generation process. This is more commonly referred to as an "actor"; however, this terminology may be confounding with language in RL, Dyna, or multiagent learning.

- *Warm-Start Background Planning*: An additional $10\,000$ updates are performed on exclusively planned experiences before play in the real game occurs.

- *Concurrent Background Planning*: Each mini-batch sampled after warm-starting contains $25\,\%$ planned experiences, and $75\,\%$ real experiences.

- *Decision-Time Planning*: In the training arenas (those that have the real game, and are not used for evaluation), the agent selects actions with a beam-search of width 3 and depth 1.

Background planning also requires defining a *search control* procedure (Sutton, 1990; 1991; Sutton & Barto, 2018). Search control defines how the agent prioritizes selecting starting states and actions for background planning. This work considers the simplest search-control method: maintain a buffer of the initial states and uniformly sample.

## C.2   World Model Implementation & Training

### C.2.1   Action-Conditioned Scheduled Sampling

As noted by Talvitie (2014), rolling out trajectories with an imperfect model tends to result in compounding errors in prediction. Their work suggests training a Markovian world model with previous predictions (referred to as "hallucinated replay"), to train the model to correct errors. For stateful world models, as studied in this work, it has been demonstrated that curricula of $n$-step future predictions can train a fruitful world model (Michalski et al., 2014; Oh et al., 2015; Chiappa et al., 2017). All of the preceding work was studying single-agent systems;

---

**Algorithm 1:** Action-Conditioned Scheduled Sampling

---

$m \leftarrow$ Initial recurrent state
**for** $t \in T$ **do**
$\quad \boldsymbol{o} \leftarrow \boldsymbol{o}^t$ if $\text{Unif}[0, 1] < \epsilon(t)$ else $\hat{\boldsymbol{o}}^t$
$\quad \hat{\boldsymbol{o}}^{t+1}, \hat{\boldsymbol{r}}^{t+1}, m \leftarrow w(\boldsymbol{o}, \boldsymbol{a}^t, m)$
**Output:** Predicted trajectory $(\hat{\boldsymbol{o}}^{0:T}, \hat{\boldsymbol{r}}^{0:T})$

---

therefore, they could assume a much more stable data distribution for training. As a result, these fixed curricula style approaches may prove fatal as the data distribution may change dramatically throughout training based on the coplayers' strategies.

Instead, this work adapts the scheduled sampling (Bengio et al., 2015) algorithm as a stochastic curricula, which will allow both short- and long-term predictions throughout the course of training. Scheduled sampling is an algorithm for training auto-regressive sequence prediction models where at each predictive step during training the model input is sampled from either the previous prediction or the ground truth. Adapting this algorithm for world model rollouts requires biasing each predictive step with the true actions while sampling between the predicted successor observation and the true successor observation. Therefore, the predictions will always be biased on true actions, but must learn to handle model-predicted observation. The sampling follows a schedule $\epsilon : \mathbb{Z} \to [0, 1]$ that determines the probability of sampling the true observation over the previous prediction. When $\epsilon$ is 1.0, the algorithm behaves akin to teacher forcing (Williams & Zipser, 1989) (with the same action-conditional modification); whereas, as it approaches 0.0 it becomes fully auto-regressive.

### C.2.2   Implementation

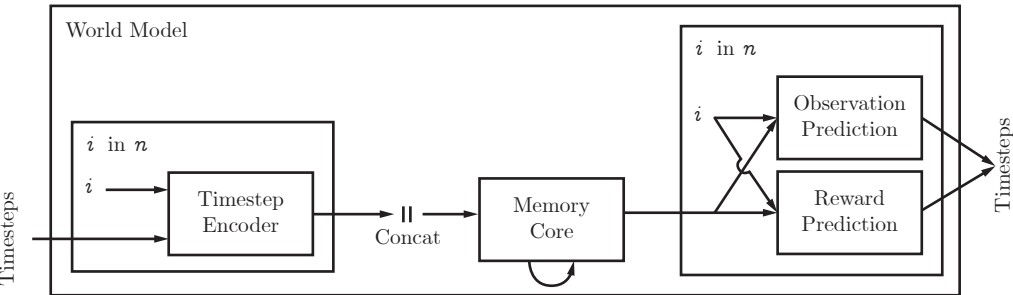

Figure 7: World Model Architecture.

The high-level architecture of the world model is illustrated in Figure 7. The world model is composed of several modules that are quite similar to the policy:

- *Timestep Encoder*: Processes all of the current observation's information into a single embedding vector. The timestep includes all new observational data that the agent gains at the current point in time. Different from the agent's timestep encoder, this encoder also receives the ID that corresponds with the timestep.

- *Memory Core*: The component of the agent that maintains and update's the agent's memory. Different from the agent's timestep encoder, this memory core receives the representation of each player's timestep concatenated.

- *Observation Prediction (Head)*: Predicts the successor observation for each player. As all games considered in this work are gridworld games, the predicted observation is a classification task for each future grid cell (that are within the respective player's observation window).

- *Reward Prediction (Head)*: Predicts the reward received for each player. Rewards are treated as categorical values.

Note, that the timestep encoder, observation prediction head, and reward prediction head each use the same parameters across each player. Similar to the agent, all components are simultaneously trained and their joint parameters are referred to as $\theta^w \in \Theta^w$. Both observation and reward losses are optimized with a cross entropy objective, and averaged across players. The total world model loss is as follows:

$$\mathcal{L}_w = \lambda_{\text{observation}} \cdot \mathcal{L}_{\text{observation}} + \lambda_{\text{reward}} \cdot \mathcal{L}_{\text{reward}}.$$

The implementation of each component is as follows:

- *Timestep Encoder*: The same as the agent's timestep encoder, but the player's ID is also provided alongside the action into the second neural network.

- *Memory Core*: Identical to the agent.

- *Observation Prediction (Head)*: The observation prediction is based on the memory core's output and a one-hot ID of the predicted player's ID. These inputs are concatenated and fed into an transposed version of the timestep encoder.

- *Reward Prediction (Head)*: A linear layer of size one. For Harvest: Categorical this output is handled as a discrete prediction; whereas, it is continuous for the other games.

A world model is trained for $1\,250\,000$ updates. Each example in the mini-batch is a sequence of $20$ transitions, where the first $5$ timesteps are used to burn-in the memory. Burn-in does not occur for examples where the first $5$ transitions are at the beginning of the episode. Moreover, sequences are added into the replay buffer at a period of $14$ so that all timesteps show up as prediction targets.

The world model is trained using action-conditioned scheduled sampling (Appendix 1, Algorithm 1). The schedule $\epsilon$ follows the following schedule:

$$\epsilon(t) = \begin{cases} 1.0 & t < 250000 \\ \frac{4}{3} - \frac{t}{750000} & 250000 \leq t \leq 1000000 \\ 0.0 & t > 1000000. \end{cases}$$

This schedule starts out training as a variation of teacher forcing (Williams & Zipser, 1989), and slowly transitions to fully auto-regressive. Additional hyperparameters are specified in Table 2.

### C.3 STRATEGIC DIVERSITY

Learning a general world model assumes that the transitions are drawn from the space of all possible transitions. This is typically not tractable, but instead draws are taken from a dataset generated from play of a *behavioral [joint] strategy* $\boldsymbol{\sigma}$. And the performance of the world model is measured under a *target [joint] strategy* $\boldsymbol{\sigma}^*$, instead of all possible strategies $\boldsymbol{\Sigma}$. Differences between $\boldsymbol{\sigma}$ and $\boldsymbol{\sigma}^*$ present challenges in learning an effective world model.

We call the probability of drawing a state-action pair $\boldsymbol{s}$, $\boldsymbol{a}$ under some joint strategy its *reach probability* $\eta^{\hat{\boldsymbol{\sigma}}}$ under joint strategy $\hat{\boldsymbol{\sigma}}$. From this, we define *strategic diversity* as the distribution

Table 2: World model hyperparameters per game.

| Hyperparameter | Harvest: Categorical | Harvest: RGB | Running with Scissors |
|---|---|---|---|
| $\lambda_{\text{observation}}$ | 1.0 | 1.0 | 1.0 |
| $\lambda_{\text{reward}}$ | 10.0 | 0.01 | 0.01 |
| Optimizer | Adam | Adam | Adam |
| Learning Rate | 3e−4 | 3e−4 | 3e−4 |
| Max Grad Norm | 10.0 | 10.0 | 10.0 |
| Batch Size | 32 | 24 | 24 |

induced from reach probabilities. These terms allow us to observe two challenges for learning world models.

First, the diversity of the behavioral strategy *cover* the target joint strategy's diversity:

$$\eta^{\boldsymbol{\sigma}^*}(\boldsymbol{s}, \boldsymbol{a}) \to \eta^{\boldsymbol{\sigma}}(\boldsymbol{s}, \boldsymbol{a}). \tag{1}$$

Otherwise, transitions will be absent from the training data. As an aside, it is possible to construct a weaker claim for coverage. This is done through making additional assumptions about the generalization capacity of a world model across transitions. For example, if transitions are drawn from two discrete latent variables, unseen combinations of these variables may be generalized if the individual values are known. However, generalization cannot be generally guaranteed, so we consider coverage.

The second challenge is that the *closer* the diversities are, the more accurate the learning objective will be. In other words, we want

$$\eta^{\boldsymbol{\sigma}^*}(\boldsymbol{s}, \boldsymbol{a}) \approx \eta^{\boldsymbol{\sigma}}(\boldsymbol{s}, \boldsymbol{a}). \tag{2}$$

If closeness is not ensured, crucial dynamics knowledge may not be learned as the learning signal is dominated from unimportant transitions. An example of the issue of closeness can be seen in the "noisy TV problem," (Burda et al., 2019). This exploration problem poses that novelty-seeking agents may be stuck forever watching the ever new TV static, and not experiencing practical novelty. In the same vein, if a world model is trained almost entirely on "noisy TV"-like experiences, and as a rarely on the few salient experiences, it may never learn. Therefore, we should strive to correct the distribution of experiences to be informed by a target strategy.

By design, empirical-game building algorithms offer a means to construct the target world model objective. These algorithms require the specification of a solution concept that serves the dual roll as the target strategy for a world model. Then through an iterative process, the empirical-game constructs strategies that progressively approach the target. In turn, generating transitions that match the target world model objective.

**Claim 1.** *Dyna-PSRO produces a correct world-model objective $\eta^{\boldsymbol{\sigma}^*}$ with a best-response oracle and a correct empirical game for a game with a unique Nash Equilibrium $\boldsymbol{\sigma}^*$.*

*Proof.* Following McMahan et al. (2003), the Double Oracle algorithm will converge to a NE in the limit of enumerating the full strategy space. Let $\boldsymbol{\sigma}^0, \boldsymbol{\sigma}^1, \ldots, \boldsymbol{\sigma}^e$ be the solutions discovered for each epoch, ending at epoch $e$. Then a dataset composed of experiences generated by the current empirical game solution evolves as follows:

$$\eta^{\boldsymbol{\sigma}^0} \to \eta^{\boldsymbol{\sigma}^1} \to \ldots \to \eta^{\boldsymbol{\sigma}^e} = \eta^{\boldsymbol{\sigma}^*}. \tag{3}$$

$\square$

The previous claim contains two strong assumptions: an exact best-response oracle and error-less empirical game. These assumptions must be made, because PSRO is parameterized by its choice of response oracle and empirical game model; therefore, PSRO's convergence must be proven for each choice. Theoretically PSRO has been shown to converge to an $\epsilon$-NE, where $\epsilon$ depends on the empirical game's modelling error, to a corresponding NE in the true game (Tuyls et al., 2020; Vorobeychik, 2010). Therefore, in practice Dyna-PSRO produces $\eta^{\boldsymbol{\sigma}^e} \approx \eta^{\boldsymbol{\sigma}^*}$, which supports the weaker claim that Dyna-PSRO generally improves the quality of a world model.

It is also worth noting the connections between this analysis and MARL regimes that seek to find any solution the game. In these regimes, the priority is finding *any* performant strategy. This matches the approach taken by the majority of studies in MARL falling under paradigms such as Independent RL or Self-Play. Therefore, their target distribution is the best-response to the previous strategy $\eta^{\text{BR}(\sigma^{i-1})}$ and changes in tandem with the strategies. When no best-response can be found, then the current strategy matches the solution and the dataset correspondingly reflects this.

## C.4 DYNA-PSRO

The Dyna-PSRO builds upon PSRO (Algorithm 3) by including the co-learning of a world model. The high-level pseudocode of Dyna-PSRO is provided in Algorithm 5 an a high-level application architecture diagram is depicted in Figure 8. There are three main co-routines of Dyna-PSRO: response computation, world-model learning, and empirical-game simulation. The details of each routine are first provided; then, how the routines interact with each other is explained.

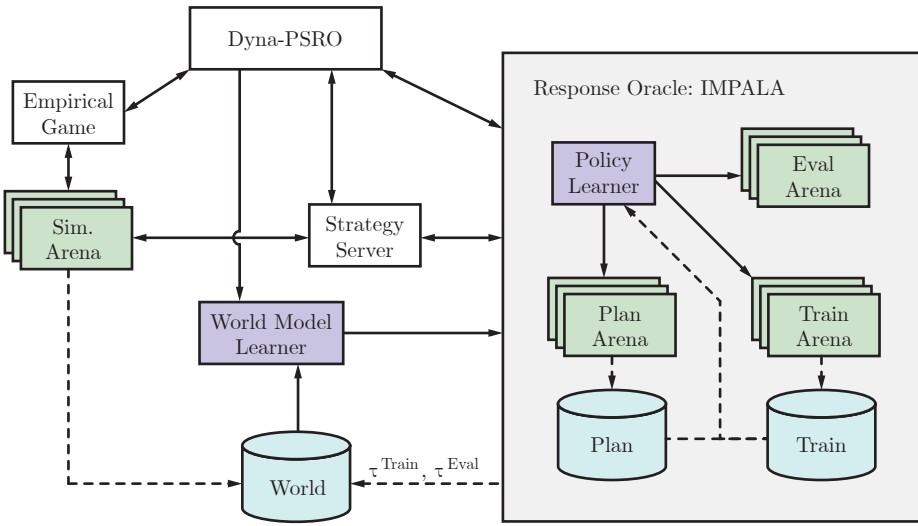

Figure 8: Overview of the major Dyna-PSRO processes.

### C.4.1 EMPIRICAL GAME

The empirical game routine is responsible for maintaining the empirical game, including simulating new payoffs and game reasoning. New profiles are sent to *simulation (sim.) arenas* for payoff estimation in parallel. Once all profiles are estimated, the game is solved, and the solution is based to the main Dyna-PSRO process. In the experiments in this work, the chosen solution is Nash Equilibrium, and it is solved through the linear complementarity (Eaves, 1971) algorithm that is implemented by Gambit (McKelvey et al., 2016).

### C.4.2 WORLD MODEL

The world model routine is responsible for training the world model and serving its parameters. This routine's pseudocode is provided in Algorithm 2, and follows mostly the same method details as the strategic diversity experiment. The difference is that instead of there being a precomputed fixed dataset, the world model is now trained over a dynamic dataset. The dataset is represented by a replay buffer that is populated from: (1) trajectories from the simulation arena used for expanding the empirical game, and (2) trajectories from the training and evaluation arenas from the response calculation. Notably, all of this data must be generated in the standard PSRO procedure, so it collected with no additional cost. The world-model learner samples and evicts data randomly from this buffer.

---

**Algorithm 2:** World Model Learner

---

**Input:** World model $w$ and data buffer $\mathcal{B}^w$
**Input:** $n$ no. of updates (default: $\infty$).
**for** $i \in [[n]]$ **do**
    Train $w$ over $\tau \sim \mathcal{B}^w$
**Output:** $w$

---

### C.4.3 RESPONSE ORACLE

The response oracle uses the IMPALA (Espeholt et al., 2018) algorithm to compute an approximate best-response to the opponent's strategy according the the current empirical game. IMPALA uses several processes that generate experiences for the agent to train on. These process are referred to in this work as arenas. The *train arenas* generate real experiences, and the *plan arenas* generate planned experiences. If the learner is using decision-time planning they will only use it in the train arenas. A third set of arenas called *eval arenas* periodically evaluate the performance of the greedy policy and record additional metrics. The arenas attempt to synchronize all parameters at the start of each episode.

The policy learner runs for a fixed number of updates, querying the datastores for experiences to learn from. The specifics of how each policy learns is described in Appendix C.1.

### C.4.4 RUNTIME PROCEDURE

A sketch of the respective processes runtime is shown in Figure 9 As in PSRO, the main empirical-game building loop iterates between response computation and empirical-game simulation.

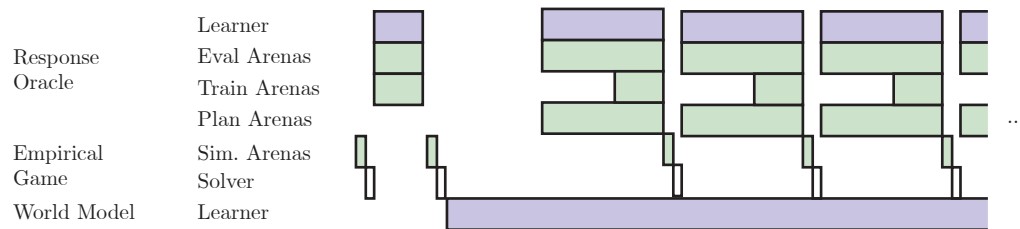

Figure 9: Example Dyna-PSRO runtime. Planning is set to occur after the first epoch. Each players' response oracle runs in parallel.

The runtime is defined by a parameter specifying on which PSRO epoch to begin planning. Before that epoch, the response oracles do not use planning at all, because the world models are untrained. All of these policies therefore are trained exclusively on real experiences just like standard PSRO. However, these experiences are also being used to populate the world model's replay buffer. Once the first planning epoch has arrived, computing responses is temporarily paused. The world model is then given a set number of updates to warm-start its parameters, before being used in response calculation. Once the world model's warm-start phase is over, all process proceed concurrently.

Throughout this work planning begins on the second epoch. The world models is given 1 million updates of warm starting.

**Algorithm 3:** Policy-Space Response Oracles (Lanctot et al., 2017)

---

**Input:** Initial strategy sets for all players $\mathbf{\Pi}^0$

Simulate utilities $\hat{U}^{\mathbf{\Pi}^0}$ for each joint $\boldsymbol{\pi}^0 \in \mathbf{\Pi}^0$

Initialize solution $\sigma_i^{*,0} = \text{Uniform}(\Pi_i^0)$

**while** *epoch e in* $\{1, 2, \dots\}$ **do**

    **for** *player* $i \in [[n]]$ **do**

        `// Algorithm 4.`

        $\pi_i^e, \_ = \text{response\_oracle}(\sigma_{-i}^{*,e-1})$

        $\Pi_i^e = \Pi_i^{e-1} \cup \{\pi_i^e\}$

    Simulate missing entries in $\hat{U}^{\mathbf{\Pi}^e}$ from $\mathbf{\Pi}^e$

    Compute a solution $\sigma^{*,e}$ from $\hat{\Gamma}^e$

**Output:** Current solution $\sigma_i^{*,e}$ for player $i$

---

**Algorithm 4:** Response Oracle

---

**Input:** Coplayer strategy profile $\sigma_{-i}$

**Input:** Num updates $k$

$\pi_i \leftarrow \theta^\pi$

$\mathcal{B} \leftarrow \{\}$     `// Replay Buffer.`

**for** *many async episodes* **do**

    $\pi_{-i} \sim \sigma_{-i}$

    $\mathcal{B} = \mathcal{B} \cup \{\tau \sim (\pi_i, \pi_{-i})\}$

**for** $i \in [[k]]$ **do**

    Train $\pi_i$ over $\tau \sim \mathcal{B}$

**Output:** $\pi_i, \mathcal{B}$

---

**Algorithm 5:** Dyna-PSRO

---

**Input:** Initial strategy sets for all players $\mathbf{\Pi}^0$

**Input:** No. of world model head-start updates $n_w$

**Input:** Epoch to begin planning $e^{\text{plan}}$

Simulate utilities $\hat{U}^{\mathbf{\Pi}^0}$ for each joint $\boldsymbol{\pi}^0 \in \mathbf{\Pi}^0$

Initialize solution $\sigma_i^{*,0} = \text{Uniform}(\Pi_i^0)$

$w \leftarrow \theta^w$

$\mathcal{B}^w \leftarrow \{\}$         `// World Model's Replay Buffer.`

**while** *epoch e in* $\{1, 2, \dots\}$ **do**

    **for** *player* $i \in [[n]]$ **do**

        **if** $e > e^{\text{plan}}$ **then**

            $\pi_i^e, \tau = \text{async}(\text{planner\_oracle}(\sigma_{-i}^{*,e-1}, w))$     `// Algorithm 6.`

        **else**

            $\pi_i^e, \tau = \text{async}(\text{response\_oracle}(\sigma_{-i}^{*,e-1}))$     `// Algorithm 4.`

        $\mathcal{B}^w = \mathcal{B}^w \cup \{\tau\}$

        $\Pi_i^e = \Pi_i^{e-1} \cup \{\pi_i^e\}$

    Wait on all futures $\boldsymbol{\pi}^e, \tau$

    Simulate missing entries in $\hat{U}^{\mathbf{\Pi}^e}$ from $\mathbf{\Pi}^e$

    Add $\tau$ from simulation to $\mathcal{B}^w$

    Compute a solution $\boldsymbol{\sigma}^{*,e}$ from $\hat{\Gamma}^e$

    **if** $e == 1$ **then**

        $w = \text{world\_model\_learner}(w, n_w)$     `// Algorithm 2.`

    $w = \text{async}(\text{world\_model\_learner}(w))$ `// Parameters periodically sync.`

**Output:** Current solution $\sigma_i^{*,e}$ for player $i$

---

---

**Algorithm 6:** Planner Oracle

---

**Input:** Coplayer strategy profile $\sigma_{-i}$
**Input:** World model $w$, real game dynamics $p$
**Input:** Warm-start background planning updates $n^{\text{BG:WS}}$
**Input:** Training updates $n$
**Input:** Concurrent background planning fraction $f^{\text{BG:C}}$
$\pi_i \leftarrow \theta^\pi$
$\mathcal{B}^{\text{plan}} \leftarrow \{\}$                // Replay Buffer with planned experience.
$\mathcal{B}^{\text{train}} \leftarrow \{\}$                // Replay Buffer with real experience.

// Asynchronously generate data on arenas.
**for** *many async episodes* **do**
    $\pi_{-i} \sim \sigma_{-i}$
    $\mathcal{B}^{\text{plan}} = \mathcal{B}^{\text{plan}} \cup \{\tau \sim (\pi_i, \pi_{-i}, w)\}$
**for** *many async episodes* **do**
    $\pi_{-i} \sim \sigma_{-i}$
    $\mathcal{B}^{\text{train}} = \mathcal{B}^{\text{train}} \cup \{\tau \sim (\pi_i, \pi_{-i}, p)\}$

// Train the response policy.
**for** $i \in [[n^{\text{BG:WS}}]]$ **do**
    Train $\pi_i$ over $\tau \sim \mathcal{B}^{\text{plan}}$
**for** $i \in [[n]]$ **do**
    Train $\pi_i$ over $\tau \sim \{(1.0 - f^{\text{BG:C}}) \cdot \mathcal{B}^{\text{train}}\} \cup \{f^{\text{BG:C}} \cdot \mathcal{B}^{\text{plan}}\}$
**Output:** $\pi_i, \mathcal{B}$

---

## C.5   COMBINED-GAME REGRET

Combined-game regret is an approximate measure of regret that all available estimates to approximate the regret within the true game. Intuitively, combined-game regret is the regret of a strategy with respect to all discovered policies. When comparing empirical-game building algorithms this is formalized as follows:

$$\text{SumRegret}(\boldsymbol{\sigma}, \overline{\boldsymbol{\Pi}}) = \sum_{i \in n} \max_{\pi_i \in \overline{\Pi}_i} \hat{U}_i(\pi_i, \sigma_{-i}) - \hat{U}_i(\sigma_i, \sigma_{-i}), \qquad \overline{\boldsymbol{\Pi}}_i \equiv \bigcup_{\text{method}} \hat{\boldsymbol{\Pi}}_i^{\text{method}}, \qquad (4)$$

where $\hat{\Pi}$ is the restricted strategy set from one of the algorithms.

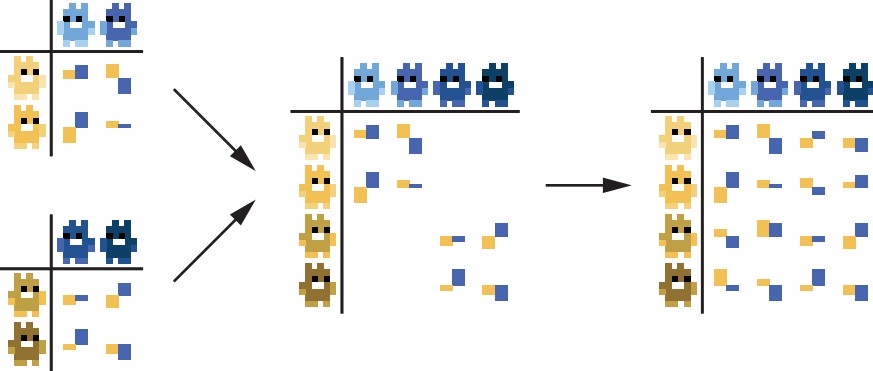

Figure 10: Combined-game construction. Left: Constituent empirical games. Middle: Combination of the strategy sets and payoff functions. Right: Completion of the empirical game by estimating new strategy profile payoffs.

The process of constructing a combined-game is illustrated in Figure 10. Where, the strategy sets (depicted by the toons) across methods are first combined. The new *combined game* that results from

this can be initialized with the payoff estimates from the constituent empirical games. Unestimated payoffs must then be simulated for the new strategy profiles. Then the complete combined game can be used to compute the combined-game regret from the solutions computed throughout the empirical-game building algorithms.

## D  GAMES

### D.1  HARVEST: CATEGORICAL

In Harvest, players move around an orchard picking apples. The challenging commons element is that apple regrowth rate is proportional to nearby apples, so that socially optimum behavior would entail managed harvesting. Self-interested agents capture only part of the benefit of optimal growth, thus non-cooperative equilibria tend to exhibit collective over-harvesting. The game has established roots in human-behavioral studies (Janssen et al., 2010) and in agent-based modeling of emergent behavior (Pérolat et al., 2017; Leibo et al., 2017; 2021).

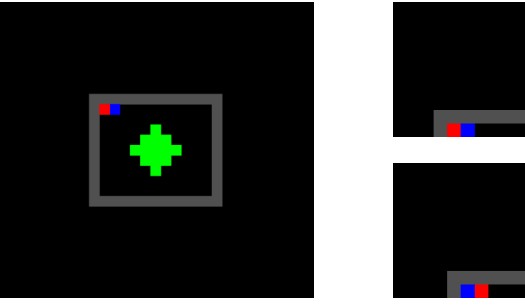

Figure 11: Harvest: Categorical. Left: game state. Right: player observations.

For our initial experiments, we use a symmetric two-player version of the game, where in-game entities are represented categorically (HumanCompatibleAI, 2019). This categorical representation facilitates faster experimentation and simplifies the interpretation of results. Figure 11 depicts the game state and player observations. Each player has a $10 \times 10$ viewbox within their field of vision. The cells of the grid world can be occupied by either agent shown in red and blue, the apples shown in green, or a wall in gray. The possible actions include moving in the four cardinal directions, rotating either way, tagging, or remaining idle. A successful tag temporarily removes the other player from the game, but can only be done to other nearby players. Players receive a reward of 1 for each apple picked. Episodes are limited to 100 timesteps.

### D.2  HARVEST: RGB

Harvest: RGB is a different implementation of the Harvest game introduced by Harvest: Categorical (Appendix D.1). Harvest: RGB is exactly the harvest implementation from MeltingPot (Leibo et al., 2021) with the same orchard map. A rendering of the game state and observations is shown in Figure 12. The main difference between the Harvest versions is that the observations are $88 \times 88 \times 3$ images of the $11 \times 11$ viewbox in front of them. There are also minor differences in the implementation of tagging and apple respawn mechanism. Episodes play for 1000 timesteps.

### D.3  RUNNING WITH SCISSORS

Running With Scissors (RWS) is a temporally extended version of rock-paper-scissors (RPS). In it, players collect rock, paper, and scissor items into their inventory. At any point the player has the option to tag their opponent if they're nearby. Then they play RPS corresponding to the distribution of items in their inventories. The agents have the same action space as in the previous games. The observation space is $40 \times 40 \times 3$ image-based viewbox in fromt of them corresponding to a $6 \times 6$ grid around them. A portion of items are placed within the game deterministically, the rest are randomly

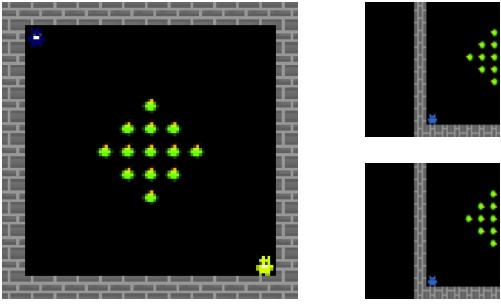

Figure 12: Harvest: RGB. Left: game state. Right: player observations.

sampled before play. If neither player tags each other before 1000 timesteps, the players are forced into playing RPS.

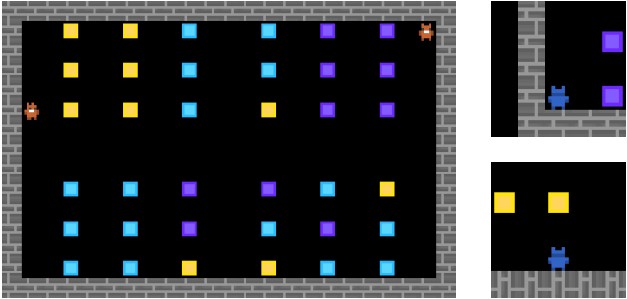

Figure 13: Running With Scissors. Left: game state. Right: player observations.

# E ADDITIONAL RESULTS

## E.1 STRATEGIC DIVERSITY

Figure 14 displays the recall results that correspond to the accuracies portrayed in Figure 2. See Section 3.1 for a discussion of these results. Figure 15 contains a simplified visualization of the results from Section 3.1.

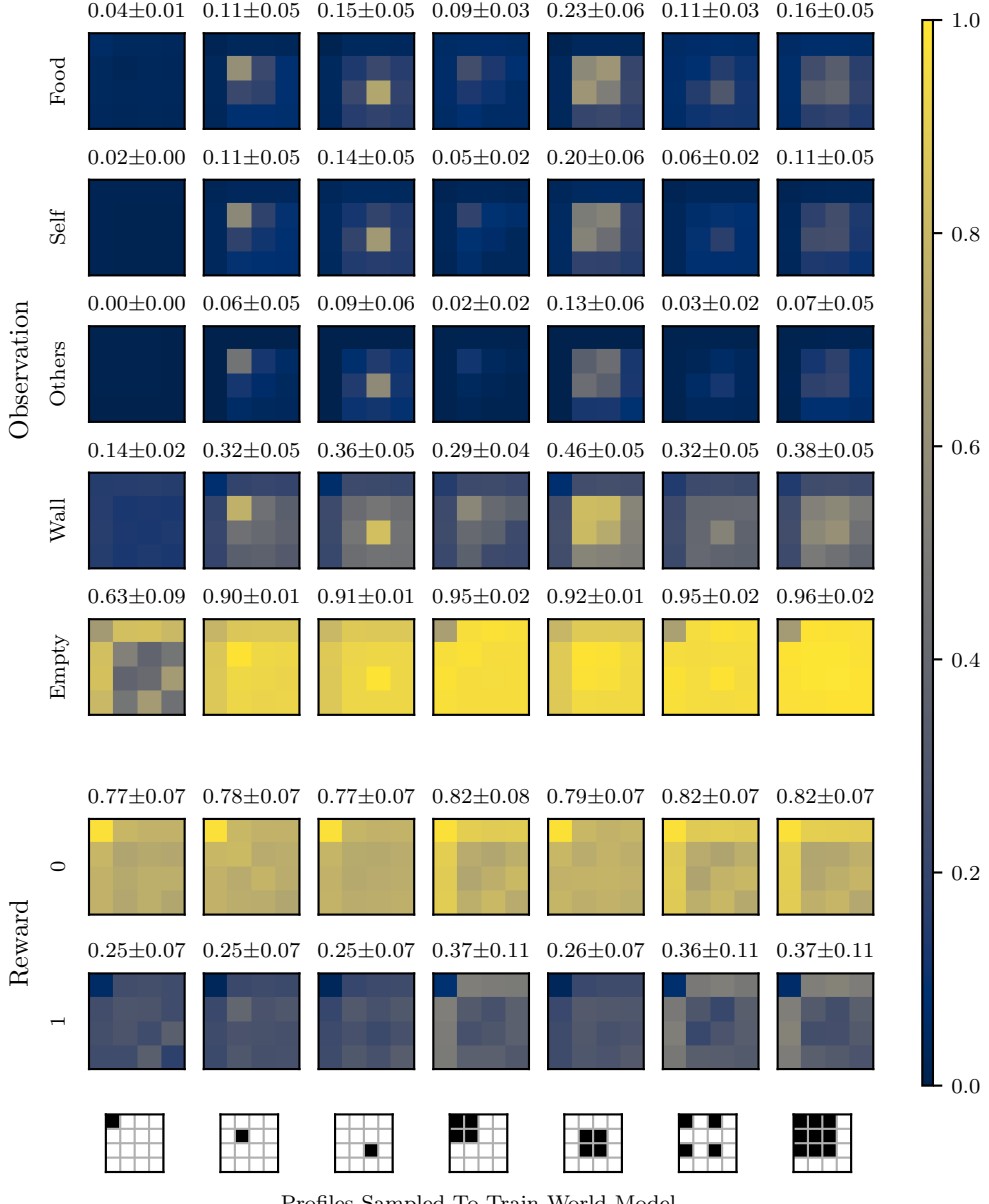

Figure 14: World model recall on Harvest: Categorical.

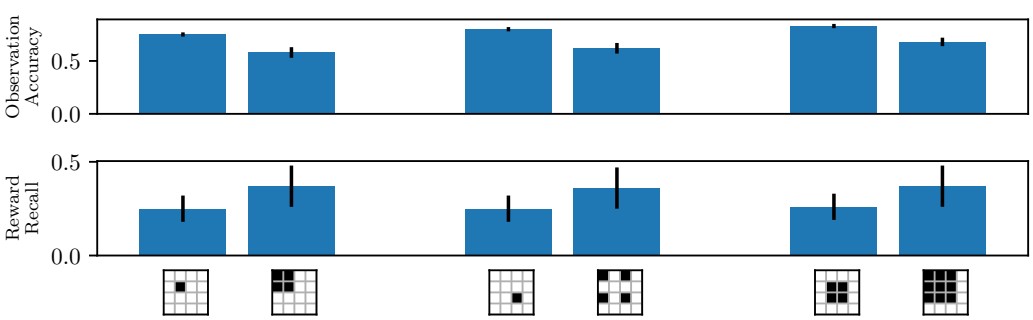

Figure 15: Impact of inclusion of random policy on world model performance.

## E.2 BACKGROUND PLANNING

Figure 16 shows the results of repeating the background planning experiment with world model ▦. Besides changing the world model, the methodology is consistent with that described in Section 3.2.1. This result shows the planner achieving results comparable to the baseline method. Supporting the adoption of planning, as it tends to not negatively impact the learning process.

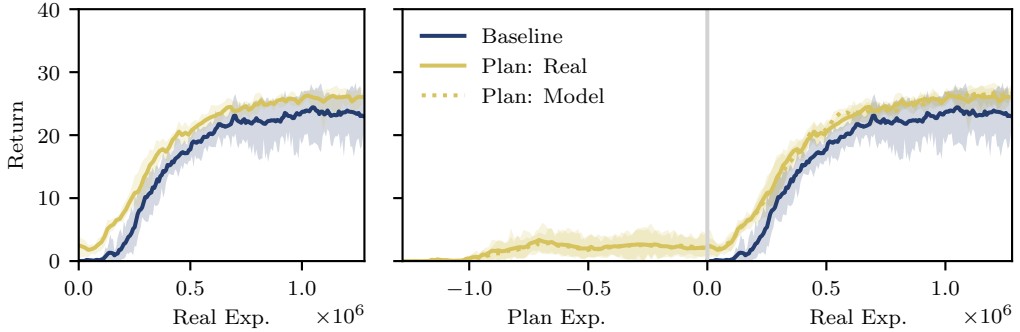

Figure 16: Effects of background planning on response computation using world model ▦. (5 seeds, with 95 % bootstrapped CI).

Figure 17 shows the results of performing both BG: W and BG: C without DT. Without DT, we observed no measurable benefit of including BG: C. As the proportion of planned experience increases in BG: C this corresponded to a decrease in the performance. We speculate that this is because the planner better fits its policy to interact with the world model instead of the real game.

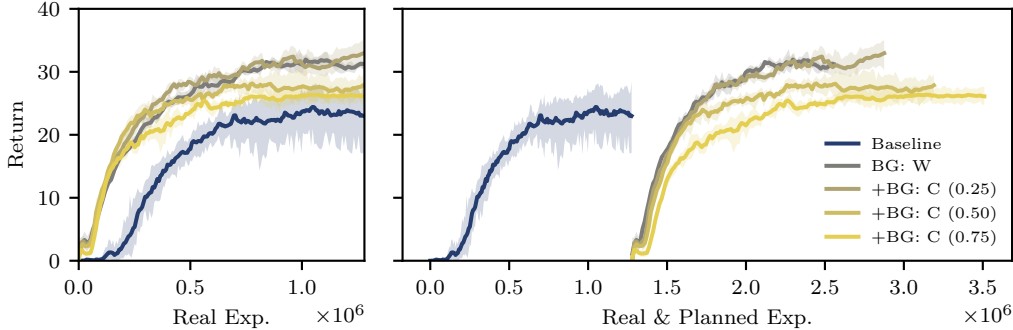

Figure 17: Effects of concurrent background planning on response computation using world model ▦. Methods labelled "+BG: C (X)" perform both BG: W and BG: C, where X denotes the proportion of planned experience within in batch of data. (5 seeds with 95 % bootstrapped CI).

## E.3 DECISION-TIME PLANNING

Figure 18 shows the results of repeating the decision-time planning experiment with world model ▦. Besides changing the world model, the methodology is consistent with that described in Section 3.2.2. This result further exemplifies the trend shown in Figure 4, where the planners that did not use BG: W failed to learn an effective policy. The planner that used BG: W achieved performance comparable to the baseline. Finally, the planner that used both BG: W and BG: C achieves the strongest performance at $33.07 \pm 6.76$. These results support the benefit of BG: W when using DT, and that effective planning performs as least as good as the baseline.

Finally, we completed an ancillary experiment to determine if planning allowed the learner to escape a locally optimal policy. To measure this we simply continued training the baseline on a significantly larger data budget to measure if its performance would improve, and even match that of the planner.

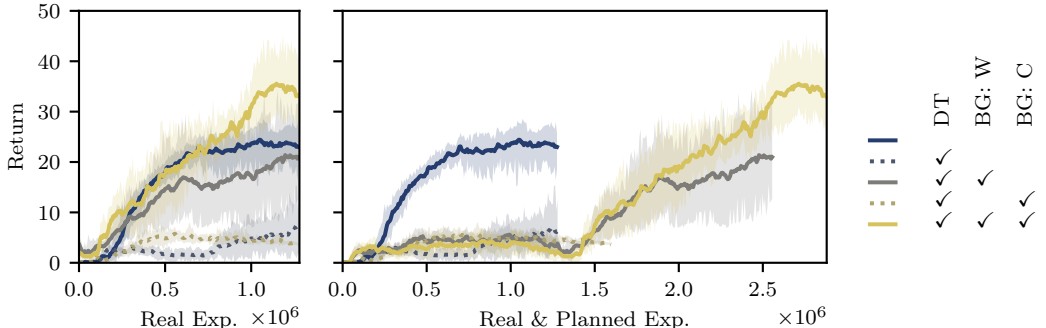

Figure 18: Effects of decision-time planning on response computation using world model ▦ . (5 seeds, with 95 % bootstrapped CI).

In Figure 19 we plot our results, and found that DT planning helped learner a stronger policy than a learner without planning.

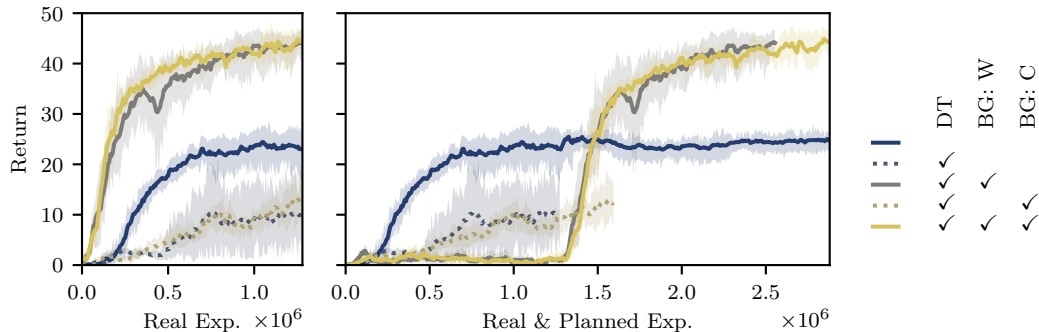

Figure 19: Decision-time planning using ▦ compared against a baseline trained longer. (5 seeds, with 95 % bootstrapped CI.)

### E.4 WORLD MODELS AS EMPIRICAL GAMES

In this section, we verify the need for separate models.

First, consider if an empirical game can substitute for a world model. The majority of previous work on empirical games represents the model in the normal form. This representation abstracts away any notion of dynamics within an episode into a choice in policy and the resulting payoff. Since empirical games currently lack dynamics information completely, this supports the choice of separate models. This is not without any exceptions. If the original game is one-shot and stateless (i.e., an episode is played through a single action), then a normal-form empirical game is exactly a world model.

Now, consider if a world model can substitute for an empirical game. World models predict successor states and rewards; and thus, can rollout planned trajectories to estimate payoffs. Note, that rolling out a trajectory with a world model is an auto-regressive prediction that tends to result in compounding errors (Talvitie, 2014; Holland et al., 2018). Despite this, it is plausible that a world model can substitute as a high-fidelity empirical game.

Figure 20 compares an empirical game estimated from real game payouts empirical games estimated with payouts predicted by a world model. In this experiment, the world models are the same that were used in Section 3.1. In general, the empirical games estimated by world models have large errors (L2 >100), with several having exceptionally large errors (L2 >1000). These result suggest that this direction may be possible with future algorithmic improvements; however, currently, the prediction

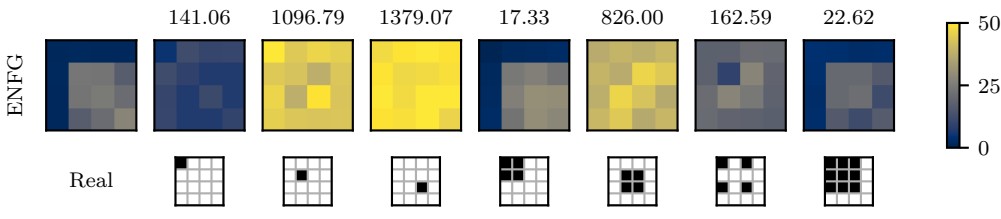

Figure 20: Empirical normal-form games (ENFG) estimated by world model rollouts. The title of each plot is its L2 distance with the real ENFG.

errors are too large to substitute empirical games with world models. Especially in games with long time horizons.