# OpenReview forum: "Co-Learning Empirical Games & World Models"
_ICLR.cc/2024/Conference — Submitted to ICLR 2024_

### Official Review · Reviewer_c2GE · 2023-10-30

**Soundness:** 3 good
**Presentation:** 3 good
**Contribution:** 3 good
**Rating:** 6
**Confidence:** 4

**Summary:**

This paper introduces a new PSRO variant called Dyna-PSRO which combines the game-theoretic framework of PSRO that trains a population of policies over an environment, with a world model that allows for transfer of training information across the population.

**Strengths:**

- In my opinion the paper addresses a valid problem in the PSRO literature. The need to re-learn policies from scratch in PSRO is an efficiency bottleneck and there exist other works that have also tried to tackle this problem (i.e. NeuPL). Therefore, because PSRO generates a large amount of training data, it makes sense to try and re-use this data in a smart way in order to generate new policies.
- Overall, the paper is well-presented and well-executed. It is generally well written and the experiment section is extensive and well-thought out in presenting the benefits of Dyna-PSRO.

**Weaknesses:**

- The primary concern I have with the paper is in relation to the discussion of strategic diversity. This concern is two-fold:
    1) There exists an extensive array of literature related to diversity within the PSRO framework which has not been discussed by the paper. For example [1][2][3][4][5]. In these works multiple diversity measures are recruited to measure the diversity over a PSRO population, and I would be interested if the authors could comment on specifically why they have chosen their variant of diversity measure over the others in the literature.
    2) The paper seems to suggest that strategic diversity amongst the population is important. Therefore, why do the authors not perform any form of strategic diversity optimisation, in order to make the population policies more diverse than if they are learned in the classic PSRO way? For example, there are no results in the work that show that there exists much strategic diversity amongst the PSRO policies, and it is quite common from my own experience with PSRO that individual policies can end up strategically very similar without any diversity optimisation.

References:

[1] Policy Space Diversity for Non-Transitive Games - Yao et al. 2023

[2] Open-ended learning in symmetric zero-sum games - Balduzzi et al. 2019

[3] Modelling behavioural diversity for learning in open-ended games - Perez-Nieves et al. 2021

[4] Towards unifying behavioural and response diversity for open-ended learning in zero-sum games - Liu et al. 2021

[5] A unified diversity measure for multi agent reinforcement learning - Liu et al. 2022

**Questions:**

It would be great if the authors could respond to the points that I have raised in the weaknesses section. Primarily:

1) Strategic diversity vs. PSRO diversity metrics

2) Strategic diversity optimisation

3) Strategic diversity evidence

I will be happy to raise my score if the authors can address these, as I think generally this is a strong paper.

---

> ### Author Response · Authors · 2023-11-12
>
> Thank you for your kind words on the quality of the presentation and execution of this work. We address your questions about strategic diversity, paraphrased here, below.
>
> > “Why have the authors chosen their measurement of strategic diversity over related works?”
>
> Our reference to “strategic diversity” is from the perspective of learning a world model. As this is a well-defined supervised learning problem, for our purposes strategic diversity amounts to wanting the training dataset to reasonably cover the test dataset. We do not mean to suggest that we are introducing a new method of measuring diversity. Instead, we’re merely framing and motivating it from the perspective of world model learning in a multiagent system. Hence, we do not claim this as a contribution nor include an extensive literature review on this topic.
>
> The reviewer offers several citations related to how considering strategic diversity may aid in strategy exploration (the process of selecting which new policy to include within an empirical game). Often these metrics of diversity ask how different is a candidate policy from a set of fixed policies. These metrics can be over the policies themselves (behavioral diversity), or how the policy strategically relates to the set of policies (response diversity). The cited works sometimes adopt ostensibly the same definition of diversity (at least in part) when they refer to behavioral diversity. The quality of these metrics is evaluated by how well they aid in improving the quality of an empirical game.
>
> The use of strategic diversity in the cited work is thus orthogonal to the consideration of world-model learning addressed here. This raises an interesting question: what is the impact of Dyna-PSRO when augmented to include diversity in its strategy exploration subroutine? We did not consider that here as  Dyna-PSRO is already a rather complex algorithm, and varying strategy exploration approaches would have complicated evaluation of our primary hypotheses.
>
> Speculating, one would expect to see that the Dyna-PSRO+diversity methods variant would continue to be experientially cheaper than PSRO+diversity due to its inherent transfer learning. However, we expect that the difference between Dyna-PSRO+diversity and PSRO+diversity to be smaller than Dyna-PSRO and PSRO. This is because +diversity methods are using their experiential budget more effectively on game solving. This raises the related question as to how these diversity metrics aid in learning world models, and how well the world models trained through Dyna-PSR+diversity generalize when compared to Dyna-PSRO. These questions should be investigated in future work, but notably don’t undermine any results within this manuscript.
>
> > “Why did the authors not try and optimize/increase the diversity of the policy set used to train the world models?”
>
> This is a fair question. We chose to not optimize for diversity in our “strategic diversity” experiments and instead chose to randomly sample policies from runs of PSRO. We did this to better emulate a “practical comparison” that we would expect to occur within a run of PSRO. Recall, our primary interest is making algorithmic design decisions that would benefit Dyna-PSRO from this isolated experiment. This method contrasts with that of an artificially constructed set of diverse policies. Any results gained from these policies may not be realized within a run of PSRO as the optimized diversity serves as a potentially large confounder. We would speculate that the benefits of diversity would be greatly exacerbated by optimizing for it within the set of policies used to train the world model. How these results would translate to Dyna-PSRO would require further experiments and would likely depend on augmenting Dyna-PSRO with diversity in its strategy exploration method (see Q1).

---

> > ### Author Response · Authors · 2023-11-12
> >
> > > “Did the authors conduct any experiments to validate the diversity of the policies used to train the world models?”
> >
> > We did, and indeed we should have included this analysis in the manuscript. We have added two comparisons of the set of policies to the appendix. The method we use to compare policies is by their action agreement. To do this we collect 30 episodes of each combination of the policies. Then we check the proportion of actions that each pair of policies agrees on across all observations.
> >
> > The results across all episodes are as follows:
> > |Policy ID|0     |     1|     2|     3|
> > |:--      |--:   |--:   |--:   |--:   |
> > |0        |1.0000|0.1262|0.1277|0.1279|
> > |1        |      |1.0000|0.8868|0.8269|
> > |2        |      |      |1.0000|0.8961|
> > |3        |      |      |      |1.0000|
> >
> > The results across all episodes except those containing the random policy are below.
> > This comparison is meant to highlight the similarity under more strategically salient episodes.
> > |Policy ID|0     |     1|     2|     3|
> > |:--      |--:   |--:   |--:   |--:   |
> > |0        |1.0000|0.1265|0.1272|0.1283|
> > |1        |      |1.0000|0.8542|0.7671|
> > |2        |      |      |1.0000|0.8608|
> > |3        |      |      |      |1.0000|

---

### Official Review · Reviewer_Y4nV · 2023-10-31

**Soundness:** 2 fair
**Presentation:** 2 fair
**Contribution:** 3 good
**Rating:** 5
**Confidence:** 4

**Summary:**

This paper explores the benefits of co-learning *transition dynamics and
expected reward signal* (using "world models") and *game models from which the
best response is inferred from strategy profiles of other players and estimated
payoff matrices* (using "empirical games"). A few experiments are run to
indicate that these paradigms can benefit from integration with each other, and
a final set of experiments demonstrate performance increases when incorporating
co-learned aspects of world-model and empirical games to a baseline model.

**Strengths:**

This paper provides an original contribution to learning in repeated games by
combining ideas in "world-models" and "empirical games", and provides experimental results.

While unexplored by the paper, the algorithmic elements proposed, by estimating
payoffs for *other* players, also has relevance to inverse game theory.

**Weaknesses:**

The empirical results of the paper could have benefited from a clearer
exposition, and some of these results are not as strong as claimed.


## Exposition
Stylistically, each experiment should be justified with a clear hypothesis in
mind, rather relying on the reader to reverse-engineer the hypothesis and how
the experiments support the central claims.

My own reverse-engineering is as follows: The experiments are motivated as a way
do demonstrate the benefits of co-learning world models and empirical games.
This is intuitively justified by:
1. The benefits to the world model anticipated by increased exploration
   (strategy diversity induced by the empirical game model). [Figure 2]
2. The benefits to the empirical game model in calculating best responses when
   allowed access to a world model. [Figures 3, 4]

## Claim 1

### Figure 2
A world-model is trained from game trajectories generated by random play or
PSRO-generated policies with restricted strategy spaces. This is reasonable for
assessing Claim 1, above, but the results are not as convincing as one would
hope (a clear trend between diversity of strategy-space samples and accuracy is
not established, nor are clear trends regarding accuracy *restricted to the
strategies used for training*). Moreover, the provided interpretation lacks
coherency: If class imbalance is causing problems, then perhaps simpler game and
setting should be used to demonstrate the desired claim.

As a minor point also regarding Figure 2: The way in which the results are
communicated is somewhat confusing. For example, why use the matrices to
represent combinations of strategies if we only considering symmetric strategy
profiles? Just give sets of strategies, e.g., $\\{1\\}, \\{2\\}, \\{1, 2\\}, \\{2, 3\\},
\\{1, 3\\}, \\{1, 2, 3\\}$.

## Claim 2
The experiments that address Claim 2, above attempt to do so by considering two
forms of "planning". The motivation is that the use of world-models can benefit
the selection of best-responses to estimated payoff matrices and restricted
strategy profiles (ie, empirical games), where the world-model is used to train
a reinforcement learning model to select best-responses.

### Figure 3
A pre-trained world-model (learned from samples with a restricted strategy
profile) is used to pre-train an RL model for best-response in a "planning"
phase, and this RL model is compared to a non-pre-trained baseline during real
play (Real play involves an opponent using a strategy omitted from the data seen
by the pre-trained world model). This experiment is reasonably constructed and
justifies the claim that the world model benefits the learned RL best-response
model through background planning.

### Figure 4
I do not understand the distinction between Decision-Time planning as used in
this paper and Markov-Chain Monte-Carlo (MCMC) to obtain better estimates of the
value function. Ostensibly, the RL model used for best-response planning is
already deploying the maximum-estimated-value action, no? In any case, the
figure appears to suggest that Decision-Time planning alone markedly *degrades*
performance, but this is ignored in the text. I do not see justification for the
claim that "world models offer the potential to improve response calculation
through DT planning," based on this figure. How does the figure support this claim?

As a minor point regarding Figure 4, the x-axis and alignment of the loss curves
on the right panel should be adjusted to agree with the analogous presentation
in Figure 3, as is done in the left panel.

## Main Experiments
While a marked improvement over PRSO, Dyna-PRSO could be compared against
additional baselines in an ablation study --- i.e., against models that
incorporate only the world model or only incorporate the "Dyna" RL component to
solve the empirical games component --- to more rigorously establish the claims
of the paper.

**Questions:**

Please address the questions regarding Figure 4, asked above, regarding
- The difference between DT and MCMC.
- How Figure 4 supports the claim that DT improves response calculation.

---

> ### Author Response · Authors · 2023-11-12
>
> Thank you for taking the time to review our manuscript. Below we have responded to the individual comments or questions that you have raised throughout your review.
>
> ### Summary
> Based on this summary, the reviewer may be assuming an overly narrow view of the use of empirical games. They write that empirical games are “game models from which the best response is inferred from strategy profiles of other players and estimated payoff matrices”. Empirical games are estimated models of the game of interest and may be studied and analyzed as one may do with a standard game. The PSRO algorithm uses empirical games to compute solutions (e.g., Nash equilibrium), both to drive strategy exploration and as a final result. (Best-response inference in PSRO is actually performed using dRL, outside the game model.)
>
> ### Exposition
> The reviewer then goes on to suggest improving the exposition of the text by associating with each experiment an explicit hypothesis. These hypotheses were laid in the abstract: “We investigate the potential gain from co-learning these elements: a world model for dynamics and an empirical game for strategic interactions. Empirical games drive world models toward a broader consideration of possible game dynamics induced by a diversity of strategy profiles. Conversely, world models guide empirical games to efficiently discover new strategies through planning.” It seems to us unfair to suggest that a reader must reverse engineer the hypotheses of the paper. We agree that we could and should more clearly and centrally state the hypotheses, and collate them better with experimental evidence.
>
> ### Claim 1
> > the results are not as convincing as one would hope (a clear trend between diversity of strategy-space samples and accuracy is not established, nor are clear trends regarding accuracy restricted to the strategies used for training).
>
> The results may not be as strong as we would have preferred, but we argue they are sufficiently positive to validate this as a promising direction of work. We found several trends:
> Including the random policy negatively impacted observation accuracy.
> Including the random policy positively impacted reward recall.
> The inclusion of more policies correspondingly reduces the observation loss (cross-entropy).
> 3.64±0.24, 5.33±0.89, 4.25±0.75, 1.99±0.23, 3.76±0.67, 1.83±0.24, 1.45±0.20 (following the order in the paper).
> We will include this additional result in the appendix.
> More importantly, this experiment offered insights into world model training, which we could deploy within Dyna-PSRO.
>
> > If class imbalance is causing problems, then perhaps simpler game and setting should be used to demonstrate the desired claim.
>
> We agree that had we chosen to study simpler games or force artificial diversity into our set of policies (see reviewer c2GE) that we could construct overwhelmingly positive results. We chose not to do this because we suspect that its results would not generalize to the practical cases of interest.
>
>
> ### Claim 2
> > I do not understand the distinction between Decision-Time planning as used in this paper and Markov-Chain Monte-Carlo (MCMC) to obtain better estimates of the value function.
>
> Decision-time planning is a classical reinforcement learning technique where planning is focused on a specific state to inform its action selection. Within this specific work, we take a very vanilla approach and do a width-and-depth-limited lookahead search. Given some candidate state s, we sample k actions, and use predict successor states/rewards for each of these k actions. Note that we’re not sampling stochastically from the transition dynamics in this work as we’re assuming a deterministic world model. We repeat this process for a fixed depth. The estimate of the action values from the original candidate state is then the maximum reward + bootstrap (using the value function to estimate the continued return from the final state in the search process) of each subtree of the search process.
>
> There are a number of ways that this differs from MCMC. The first is, that we are assuming deterministic transitions so have no need to sample over transition spaces. The second is that we do sample the full return in our planning procedure as one would with a Monte Carlo method, and instead leverage our value function to bootstrap this estimate.
>
> It is possible to use Monte Carlo methods for Decision-Time planning in Dyna-PSRO for correspondingly compatible classes of games. The benefit of using the world model is its predicted partial returns should offer a more accurate estimate of the value function. This does assume inaccuracies in the value function and corresponding accuracies in the world model.

---

> > ### Author Response · Authors · 2023-11-12
> >
> > > I do not see justification for the claim that "world models offer the potential to improve response calculation through DT planning," based on this figure.
> >
> > The improvements of DT planning can be seen by it computing a policy nearly twice as strong as policy without planning (Figure 4). The caveat of this result, as discussed in the paper, is that this background planning should be used in conjunction with decision-time planning. As we mention in the preceding sentence “[T]he main outcome of these experiments is the observation that _multi-faceted_ planning is unlikely to harm a response calculation, and has a potentially large benefit when applied effectively.”
> >
> >
> > ### Main Experiments
> > > Dyna-PRSO could be compared against additional baselines in an ablation study --- i.e., against models that incorporate only the world model or only incorporate the "Dyna" RL component to solve the empirical games component
> >
> > World models do not constitute game-solving algorithms and would amount to self-play/independent-RL which is well-known in the literature to exhibit poor generalization performance. Moreover, these potential baselines are not optimizing towards a solution concept, and therefore, are addressing a different problem then PSRO/Dyna-PSRO and would be unfit to compare against.
> >
> > We’re not sure what the reviewer here suggests by only “incorporating the ‘Dyna’ RL component to solve the empirical games component” Dyna is definitionally the integration of planning, learning, and acting. The closest baseline is removing planning, which is the PSRO baseline we have included.
> >
> > The experiments in the preceding sections are meant to serve as a variety of candidate baselines for Dyna-PSRO with varying degrees of planning. Running each of the planning ablations in the full game-solving context would be extremely computationally demanding. Therefore, we settled for studying them in limited settings and deploying the must successful configuration for game solving. Therefore, we believe we have offered a diversity of baselines to establish the effective components of Dyna-PSRO.

---

> ### Comment · Reviewer_Y4nV · 2023-11-16
>
> I appreciate the detailed reply to my review.
>
> Thank you for clarifying better that PSRO solves for Nash equilibria using dRL
> outside the game model, and I retract my criticism that a more thorough ablation
> study was not performed in the main experiment.
>
> My chief criticism was not the overarching hypotheses of the paper, but rather
> that it was not immediately obvious to me how each experiment's outcomes
> supported the overarching hypotheses. That is, the overarching hypothesis should
> provide a clear expectation for what results from the experiments should be (the
> experiment-specific hypothesis), which can be quickly compared to the actual
> results. As it is, I find the connection between the experimental results and
> the core hypothesis to be somewhat convoluted in presentation.
>
> Overall, I am happy to improve my rating for contribution from 2 to 3, and my
> overall rating from 3 to 5.
>
> I will respond to the provided feedback below.
>
> ### Figure 2
>
> > The random policy negatively impacted observation accuracy.
> > Including the random policy positively impacted reward recall.
>
> I now better understand the results communicated by Figure 2. I still find the
> way these results are presented to be lacking in simplicity and clarity, but I
> acknowledege that they are stronger than I initially realized.
>
> ## Claim 2
>
> > We’re not sampling stochastically from the transition dynamics in this work as we’re assuming a deterministic world model.
>
> Thank you for clarifying.
>
> ## Figure 4
>
> > The caveat of this result, as discussed in the paper, is that this background planning should be used in conjunction with decision-time planning.
>
> Let me know if this analogy is apt: DT is like liquid soap, and background-planning is like water.
>
> In a study to determine the most effective way to wash one's hands, we compare
> + rubbing dry hands together.
> + rubbing hands together under water.
> + rubbing hands together, with soap on them, under water.
> + We don't consider the use of liquid soap without water.
>
> Rather make the claim that "multi-component liquid use is unlikely to harm
> hand-washing, and may help help when applied effectively", it would be far more
> precise to claim "Results indicate that water alone can be more effective at
> cleaning hands than dry rubbing, while soap used with water does the best."
>
> It's not that the claim you've made is wrong, per-se, but it is rather weak and,
> again, strains the relationship between the experimental results and the central
> hypotheses / results you communicate.

---

> > ### Author Response · Authors · 2023-11-19
> >
> > We thank the reviewer for continuing to offer help that will improve our manuscript. We're glad that we were able to clear up some of your concerns and questions. The crux of the mentioned concerns lies in the relationship between presenting results and tying them back into the overall narrative of the work (esp., hypotheses). We agree this is an area for improvement and will continue to work towards this goal for the final version of the manuscript. The analogy in Figure 4 correctly captures one such point of improvement. The later interpretation of the result is stronger and more clear. We thank the reviewer for this helpful suggestion and will make the corresponding change. We are open to and welcome any further questions or suggestions you may have.

---

### Official Review · Reviewer_qJSG · 2023-11-02

**Soundness:** 3 good
**Presentation:** 3 good
**Contribution:** 3 good
**Rating:** 8
**Confidence:** 3

**Summary:**

In this paper, the authors consider both world dynamics and strategic interactions among agents when considering strategies for games. They explore the benefits of co-learning world models for dynamics and empirical games for strategic interactions. The authors introduces a new algorithm called Dyna-PSRO, which combines these two elements and demonstrates better performance compared to a baseline algorithm (PSRO) in partially observable general-sum games, particularly in terms of lower regret and fewer required experiences. Their approach proves advantageous in scenarios where collecting player-game interaction data is costly.

**Strengths:**

Overall, I think the idea of exploring the combined training using empirical games and world models  is an interesting idea. The paper is well-written and the design of the experiment seems very reasonable in terms of three different games.

**Weaknesses:**

This is probably the theoretician in me, but it would be nice to see some theoretical guarantees measuring how much of an improvement Dyna-PSRO is over PSRO. I understand that this might be difficult to obtain given that it might be game-dependent.

**Questions:**

Q1) I see you selected two versions of Harvest and Running with Scissors for testing. Is there any reason behind these choices?

Q2) This is an honest question: Were you surprised by the results? By combining the two (empirical game and world model) you would expect that the training should be no worse, no? However, the paper measures, at least empirically, how much does combining the two improves the SumRegret and the return?

Q3) Is there anything that can be said about how many less experiences are needed in the Dyna-PSRO vs. the standard PSRO?

Q4) What are the specifications of the system used to run the experiments?


Minor typos/comments

I'm assuming NE in the first paragraph of Section 2 refers to Nash equilibria, but it should be spelled out once for the sake of completeness.

Section 3 could be spelled out a bit better in terms of definitions. There are some variables undefined like $\Delta(\mathcal{R})$ and $\mathcal{A})$ which should be spelled out for the sake of completeness, though I know they are of common use.

---

> ### Author Response · Authors · 2023-11-12
>
> Thank you for your review and your kind words including that the paper is “well-written” and that the “design of the experiments seems very reasonable”. We address your individual comments and questions below.
>
> > “... it would be nice to see some theoretical guarantees …”
>
> We agree that theoretical results would be advantageous. However, due to the complexity of studying games in general, we were unable to construct any bounds that improved upon PSRO. This is primarily due to the convergence of learning-based game-solving algorithms hinging on using epsilon best-response subroutines. In idealized settings, one could construct games where model-based RL (Dyna-PSRO) is guaranteed to produce a smaller epsilon BR than model-free (PSRO). However, it’s not understood how these small epsilon errors contribute to the overall algorithm's long-term progress toward game-solving. This is because small epsilon noise in the best response can help PSRO explore new policies in a way that could further improve the empirical game. We agree that this is a super interesting research direction, and any progress toward it would be a great advancement.
>
> **Q1:** “How did you select the games for evaluation?”
>
> We selected these games for several reasons. The four reasons behind choosing Harvest were: (1) we had access to both simple (categorical) and more complex (RGB images) variants of the game allowing for in-depth analysis, (2) its popularity in the field of multiagent reinforcement learning, (3) we had experiences running experiments on the game, and (4) it is general-sum meaning it represents an important class of games. As for Running With Scissors, we both similarly had experience training agents in this game as well as it being a strictly competitive (zero-sum) non-transitive game. Thereby, representing a different and important class of games than Harvest.
>
> **Q2:** “Were you surprised by your results?”
>
> We were not generally surprised that Dyna-PSRO outperformed PSRO, because including transfer learning should at least preserve performance. It is possible for transfer learning to hurt performance, but because we had studied the co-benefit problems independently and extensively before deploying them in Dyna-PSRO, we had confidence that it was unlikely to hurt performance in the worst case. We were, however, surprised by how large of an improvement that was possible without employing more advanced planning algorithms.
>
> **Q3:** “Is there anything that can be said about how many fewer experiences are needed in the Dyna-PSRO vs. the standard PSRO?”
>
> This is a really good question and one we’re still thinking about. As we mentioned in Q1, the community currently doesn’t fully understand the relationship between the best-response quality and how many rounds of empirical game expansion are required. As a result, only last-iterate convergence guarantees could be readily made and they would be straightforward adoptions of guarantees from the model-based reinforcement learning literature (requiring strong assumptions about the nature of the game). Speculating, we would expect games with simpler dynamics and a larger strategic space to benefit more from Dyna-PSRO. However, games with exceptionally complex dynamics may see little benefit as learning an effective world model is difficult.
>
> **Q4:** “Specifications of the systems used in the experiments?”
>
> In Appendix B we specify the compute settings for our experiments. Copied below for convenience. “GPUs are used for training world models, and policies within Dyna-PSRO. Two types of GPUs were used throughout this work interchangeably: TITAN X and GTX 1080 Ti. All other computation was completed using CPUs. Each response calculation had additional CPUs corresponding to the number of experience generation arenas described in Appendix C. Experiments were run on internal clusters”
>
> Thank you for the editorial suggestions, we agree that fully defining these terms will help with the completeness and clarity of the work. We will incorporate these changes in the final version of the paper.

---

> > ### Comment · Reviewer_qJSG · 2023-11-16
> >
> > Thank you for addressing my concerns.

---

### Official Review · Reviewer_sm74 · 2023-11-03

**Soundness:** 2 fair
**Presentation:** 3 good
**Contribution:** 2 fair
**Rating:** 6
**Confidence:** 4

**Summary:**

In this paper the authors present an approach to simultaneously learn a world model and empirical game. The basic concept is that the empirical game benefits the world model by ensuring more diverse training data and the world model benefits the empirical game by allowing for simulated planning. Their results demonstrate that their combined approach can outperform approaches that only learn an empirical game.

**Strengths:**

The paper is quite original. To the best of my knowledge it represents the first major effort to combine world model and empirical game learning. The quality of the work in terms of describing the approach itself and the evaluation setup is also sufficient. There are no issues with the clarity of the paper in terms of explaining the core of the approach, though some details could be improved. Finally, in terms of significance, the paper can clearly improve the performance of agents attempting to play multiplayer games of a certain type, which is likely to be of interest to researchers working on empirical games or multiagent settings generally.

**Weaknesses:**

The current paper draft has one minor weakness and one major weakness from my perspective.

The minor weakness is the relative lack of clarity on the approach in the paper itself. The appendix implementation details help but there's still some things that are unclear. As an example of what I mean, the authors state: "This schedule starts out training as a variation of teacher forcing, and slowly transitions to fully auto-regressive." It's clear from this statement what is happening at a high level, but not the exact setup for how the models are trained. This kind of imprecision is unfortunately common in the current draft concerning the authors' approach.

The major weakness(es) of the paper are the results. Specifically, they do not fully seem to support the authors' stated claims. Section 3.1 seems to actually contradict the claims that empirical game learning can benefit world model learning, since the approach is outperformed in terms of reward prediction by a random sampling approach. Further, the games that are employed throughout the paper are rather simple in comparison to the common games used to evaluate world models (e.g. Pong, Doom, Pacman, Cheetah Run, etc.). I understand that the authors are focused on multiagent settings, but it might have been helpful to construct a setting more like a traditional world model environment, which would have also allowed for a comparison against a World Model baseline. As it is, my takeaway from the results are that world models benefit empirical games but not necessarily the other way around. This is still a contribution, but a more limited one with less general significance. It also does not match authors' stated claims and contributions.

**Questions:**

1. What is the process from start-to-finish of training your approach (at a high level)?
2. Do the authors disagree with my interpretation that the results do not support that empirical games benefit world models? If so, why?
3. Why not employ a more complex evaluation game?
4. Why not include a world model baseline?

---

> ### Author Response · Authors · 2023-11-12
>
> Thank you for your kind words on the originality, quality, and interest of our work. We address your comments and questions below (paraphrased).
>
> > “Lack of clarity in the approach. For example, action-conditioned scheduled sampling.” / Question 1
>
> We apologize for the lack of clarity and would be more than happy to elaborate on any specific details that any reviewer found unclear. We agree that this is an area for improvement in the manuscript as the algorithms being used are complex and challenging to convey in limited pages. Despite this, we hope that our effort to include pseudocode, hyperparameter tables, algorithmic descriptions, graphical descriptions, and a promise to release code help alleviate these concerns in the long-term. We will continue to work on improving the clarity of the text as suggested.
>
> In regards to action-conditioned schedule sampling, we believe that clarity may have been compromised due to separating details across sections C.2.1 and C.2.2 and to a missing high-level description of the World Model training process. Thank you for pointing out this missing detail. The world model is trained as a sequence-to-sequence supervised learning problem. This motivated our analogies to “teacher forcing” and “auto-regressive” training regimes. To expand, as requested in Question 1, we train our world models in the strategic diversity section on a fixed dataset. For each gradient step, we sample a mini-batch of sub-trajectories of length 20. Part (5 timesteps) of the sub-trajectories are used to burn-in/initialize the recurrent state of the world model. Notably, this means that we’re not performing backpropagation through these timesteps. For the remaining timesteps, we perform action-conditioned scheduled sampling (C.2.1, Algorithm 1) to train our model. Essentially, for the remaining timesteps we take the previous world model's predictions and randomly feed them as input to the model (auto-regressively) as opposed to using the true next timestep from the dataset. We define an epsilon-schedule (C.2.2) that describes how this random sampling occurs throughout training. At the beginning of training, we are performing teacher forcing: giving the world model ground truth inputs as it predicts the full sub-trajectory. After many updates, we start randomly not using ground-trough timesteps within a trajectory, but instead feed in the models previous output as input. By the end of training we are only giving the world model the burn-in timesteps and all inputs are auto-regressive. This allows the world model to learn to make accurate predictions given any artifacts induced by the world model. We will update the text of the paper to make this all clearer.
>
> > “Results do not fully seem to support the authors’ stated claims.” / Question 2
>
> It is not accurate to say that “random sampling approach[es]” would outperform any of the world models we trained. The class imbalance in rewards, favoring unrewarding states, means that random sampling would actually perform quite poorly in terms of reward-prediction accuracy. It is true, however, that predicting only the most frequent class, 0 reward, will register as having a very strong reward-prediction accuracy. This doesn’t indicate a problem with our trained world models, but with using accuracy to evaluate the quality of world models. These metrics don’t measure their effectiveness in the downstream task of interest: planning. We briefly mention this in Section 3.1’s discussion, but the nuance is omitted due to paper length requirements. As a stopgap to measuring planning performance in isolation, we considered further metrics such as recall and saw small positive benefits. Furthermore, in Section 3.2 and the appendix we perform planning experiments with both the {1,2,3} world model and the {1} world model. When compared (background: Fig3 v Fig16, decision-time: Fig4 v Fig19), we found that the more strategically diverse world model led to much more successful policy training. Moreover, if you examine the world models observation cross-entropy loss across the same diversity order in the paper you observe: 3.64±0.24, 5.33±0.89, 4.25±0.75, 1.99±0.23, 3.76±0.67, 1.83±0.24, 1.45±0.20. This is another instance of seeing a positive correlation between strategic diversity in world model performance. These arguments do not directly appear in the body of the paper and we will revise the manuscript to make these results clear. In total, we believe that all of this evidence adequately supports the strategic diversity claim in the paper and that empirical games benefit world models.  We would also like to stress that the strategic diversity result is meant only as a point of novelty (as mentioned in the introduction), and is not meant to be a major claim or contribution of the paper.

---

> > ### Author Response · Authors · 2023-11-12
> >
> > > “Games are simpler than standard games used to evaluate world models.” / Question 3
> >
> > We respectfully disagree that our selected games are appreciably simpler than the standard world-model games (save, Minecraft, but there is limited success in this domain). A key difficulty in our games not present in the recommended games is that they are partially observable. This adds a huge amount of complexity to training an effective world model as it must infer from a small viewbox around the agent the state of the much larger world around it. Moreover, multiagent systems cause the dynamics learned by the world model to present as nonstationary as other agents are taking unseen actions. This furthers the difficulty of the learning problem.
> >
> >
> > > Question 4: Why not include a world model baseline?
> >
> > Standard model-based reinforcement learning (MBRL) algorithms are not optimized for game solutions. Instead, they are typically trained in a way that would generate one set of performant agents that would be unable to generalize to different coplayers (a well-known issue in multiagent learning algorithms). PSRO modifies RL to optimize for game solutions through its empirical game. Therefore, a baseline MBRL algorithm without an empirical game would be an unfair comparison as it is optimized for a different objective. It would be expected to perform quite poorly, and how to directly and fairly compare the two algorithms (PSRO vs MBRL) is unclear.
> >
> > To speculate, we would expect MBRL to perform roughly as well as PSRO at ~3 epochs. This is because, at this point in PSRO, we have essentially trained a BR(Random) and BR(BR(Random)). Why we speculate this is comparative to MBRL is because of symmetry breaking. At some point when the two MBRL policies are training, one agent will learn an effective policy (roughly BR(Random)) and the other can only learn to do the best it can in this setting (roughly BR(BR(Random)). Anecdotally, this overfitting to one specific interaction protocol has been repeatedly observed in many multiagent works. We would speculate overcoming this point in training would require considering the strategic nature of the game explicitly, as PSRO most naturally does.

---

> > > ### Comment · Reviewer_sm74 · 2023-11-14
> > > **Re: Official Comment by Authors Q3+Q4**
> > >
> > > For the Q3 response, I would disagree that these games are not partial observable. This is clearly true of Pacman as modelled by NVIDIA's GameGan, but is also true of the common VizDoom and Pong environments due to stochastic elements (enemy movement, spawning, etc.). I definitely agree that the partially observable elements are more complex in the paper's environments than VizDoom and Pong, but the comparison to Pacman is less clear. Beyond the partial observability issue, I'd be curious to see how the proposed approach models complex observable dynamics present in these games. This would help determine the generalizability of the approach. Is the approach only relevant to games with complex partially observable dynamics or to other games?
> > >
> > > For Q4, the speculation in the answer is well-grounded, but still speculation. Particularly given the results comparing the random baseline to the proposed approach for background planning and accuracy, a more traditional world model would have been an appropriate additional baseline to help further clarify the results.
> > >
> > > Overall, given the change in Q1 I will update my recommendation, but only by one step as I feel there are still issues in Q2-Q4.

---

> > > > ### Author Response · Authors · 2023-11-15
> > > >
> > > > Thank you for continuing to engage with our work. We're glad our response to Q1 addresses your immediate questions. Below we address the unresolved comments and questions.
> > > >
> > > > ## Q2
> > > >
> > > > **Supporting Claims.** The reviewer is correct to suggest that planning performance does not necessarily imply that it occurred because of "a broader consideration of possible game dynamics," by the world models. Instead, we claim that the combined 5 different results together support this claim:
> > > > - Including a random policy hurts observation-prediction accuracy, because the dataset has a reduced set of game dynamics (Figure 15).
> > > > - Including a random policy helps reward-prediction accuracy, because it helps the class imbalance in rewarding data (Figure 15).
> > > > - A world model trained on more strategically diverse data (that _should_ support a broader set of game dynamics) is more performant at _background_ planning ({1}: Figure 17; {1, 2, 3}: Figure 3).
> > > > - A world model trained on more strategically diverse data (that _should_ support a broader set of game dynamics) is more performant at _decision-time_ planning ({1}: Figure 18; {1, 2, 3}: Figure 4).
> > > > - Including additional strategic policies improves the observation loss (cross-entropy) of the world model.
> > > >
> > > > | {1} | {2} | {3} | {1, 2} | {2, 3} | {1, 3} | {1, 2, 3} |
> > > > |---|---|---|---|---|---|---|
> > > > |3.64±0.24| 5.33±0.89| 4.25±0.75| 1.99±0.23| 3.76±0.67| 1.83±0.24| 1.45±0.20|
> > > >
> > > > We agree that these results are not overwhelmingly positive, but we argue that they sufficiently support the claim. Our concluding statement summarizing the totality of our results on strategic diversity reflect this strength of results: "Overall, these results provide **some evidence supporting the claim** that strategic diversity enhances the training of world models." Moreover, we have not listed this claim as a major contribution of this work. We did not list this claim as a contribution as we felt it was generally known in the community: assuming a perfect learning algorithm training on a dataset (covered transitions) that better covers the test dataset (possible transitions) supports a more general [world] model. Therefore, we did not construct a trivial example where the results were overwhelming positive, and instead chose a practical investigation.
> > > >
> > > >
> > > > **Background Planning Results.** There are two main things to consider when evaluating the performance of a transfer learning algorithm: (1) the time to convergence, and (2) the final performance. The former indicates how much sooner one could _stop_ the learning algorithm, and gain benefit (measured by number of updates) from the early stopping. This is a softer metric, because determining the exact point of convergence is often unclear. Therefore, we have only light qualitative results on this front: planner {1,2,3} starts learning earlier. The later is a more clear metric, and we would argue that the results are not "fairly comparable" as there is strict improvements in terms of average return and reduction in variance:
> > > >
> > > > |Learner | Final Return |
> > > > |---| ----|
> > > > |Baseline|23.00 ± 4.01|
> > > > |Planner {1}| 26.05 ± 1.30|
> > > > |Planner {1, 2, 3}| 31.17 ± 0.25|
> > > >
> > > >
> > > >
> > > > ## Q3
> > > >
> > > > > "I would disagree that these games are not partial[ly] observable.""
> > > >
> > > > We apologize for the miscommunication. From your original list we assumed the reviewer was primarily referring to the standard Atari benchmark (e.g., Pacman and Pong) and DeepMind control suite. We are not familiar with the details of "NVIDIA's GameGan," but agree that we mischaracterized VizDoom and apologize for that error.
> > > >
> > > >
> > > > >  I'd be curious to see how the proposed approach models complex observable dynamics present in these games. This would help determine the generalizability of the approach.
> > > >
> > > > We agree that results across a larger suite of games would better inform the performance of our specific world-modeling approach. Unfortunately, game-solving algorithms, the primary point of this study, are extremely expensive to run thus limiting the extent of our study. Note, that we do not claim to contribute any advances into the design of world models, and adopt a naïve instance here in this initial study on the topic. This is because Dyna-PSRO is already an exceptionally complex algorithm, we needed to limit complexity to further our understanding of it.
> > > >
> > > >
> > > > > Is the approach only relevant to games with complex partially observable dynamics or to other games?
> > > >
> > > > The general game-solving algorithm we contribute in this work, Dyna-PSRO, is not limited to any particular class of games nor method of world modeling. It suggests only employing world models for transfer learning during game-solving, and that we can benefit in the simplest cases. With respect to our particular world model method, it is applicable to discrete and deterministic games (including fully observable dynamics).

---

> > ### Comment · Reviewer_sm74 · 2023-11-14
> > **Re: Official Comment by Authors Q1+Q2**
> >
> > Thanks for your detailed response to Q1. That helped improve my understanding of the approach considerably, and I think this sort of walkthrough would benefit the paper.
> >
> > For Q2, I apologize for oversimplifying the situation. I understand that the proposed approach outperforms in terms of rewards prediction. I also definitely agree that there seems to be a strong improvement in terms of decision-time planning. However, the background results seem fairly comparable to me. I definitely commiserate over the difficulty of finding good metrics for this problem, but overall it seems to me that while the proposed approach improves the decision-time planning performance, it does not necessarily lead to "a broader consideration of possible game dynamics". Thus I still would argue that some of the claims are not supported.

---

### Official Review · Reviewer_SAe1 · 2023-11-05

**Soundness:** 2 fair
**Presentation:** 1 poor
**Contribution:** 2 fair
**Rating:** 3
**Confidence:** 4

**Summary:**

The authors provide a method for simultaneously learning world models (i.e. models of the transition dynamics) and empirical games (i.e. estimates of the per-player payoff). The main benefits of this are twofold: (1) by incorporating PSRO-style policy generation, one obtains a more diverse range of strategy dynamics, which then leads to more diverse data on which to train the world model and (2) by reusing a single learned world model within the PSRO training loop, one can achieve greater efficiency in subsequent generations of PSRO. The resulting algorithm (Dyna-PSRO) is shown to outperform PSRO in terms of the sum of regret against all policies generated by either method.

**Strengths:**

- The high-level motivation for this work is good. Given the recent success of world-modelling in single-agent RL (e.g. the Dreamer series), one might well imagine that world modelling can be used to good advantage for multi-agent RL. Moreover, the strategic diversity offered by PSRO may well lead to better exploration of the state space, thus aiding world model construction. The "win-win" here is intuitively sensible, and to my knowledge has not previously been investigated in the literature.

- The related work section is reasonably thorough (although some recent citations are missed, see "Weaknesses").

- The Dyna-PSRO algorithm is well-described and the results in Figure 5 are reasonably convincing, with some caveats in the "Weaknesses" section below.

**Weaknesses:**

- I find it hard to understand why PSRO is a reasonable algorithm for the Harvest game, and the authors do not provide a strong argument or qualitative evidence here. At a high-level, the Harvest game reduces to a social dilemma, and therefore the Nash solution is not the desirable one (for it means that everyone will defect at the resources will be exhausted, leading to low individual and collective return). I suspect that PSRO and Dyna-PSRO are finding a variety of different solutions that succeed in defection. Indeed, low regret against exploitative opponents will exactly correspond to defection. Instead, what one should be looking for in this context are policies which incentivize others to cooperate (along the lines of opponent modelling, for instance). Now, world modelling does, in principle, help here (see this recent paper: https://arxiv.org/pdf/2305.11358.pdf). But without clear measurements of the individual return, collective return and qualitative analysis of the policies, the reader cannot judge whether the world model yields better or worse outcomes in Harvest. This leads to a key question:

(*) Do the PSRO or Dyna-PSRO agents find cooperative solutions in Harvest? If not (as the case seems to be from the discussion on page 5), what is the argument for using PSRO / Dyna-PSRO in this environment?

- The order in which the paper is presented is confusing. The main algorithm, Dyna-PSRO, is not introduced until very late in the paper. Since this is the main result, the authors would do better to introduce this first, and then present the additional sections as ablations or analyses.

- The results do not compare to existing strong baselines. For instance there are many existing papers that produce agents with good performance on Harvest and Running with Scissors (e.g. https://arxiv.org/pdf/1906.01470.pdf, https://arxiv.org/pdf/2102.02274.pdf, https://arxiv.org/abs/1803.08884). To what extent does this new method outperform the baselines?

- The results are very hard to interpret. In Figure 5, why do the Dyna-PSRO curves terminate before the PSRO curves? In Figures 3 and 4, what is the difference between the left-hand plot and the right-hand plot? The Figure captions in Section 3.2 require significant clarification, because it is extremely hard for the reader to assess the results in this section at present.

- There are several choices in the "strategic diversity" section which seem arbitrary and for which the authors have not provided motivation. For instance, why are two PSRO policies used? What is meant by "the PSRO policies were arbitrarily sampled"? Why are the PSRO policies subsampled, and what is meant by this? Why is sampling a different policy from PSRO a good test of generalization (as opposed to having a held-out policy for generalization trained with a method from the literature, which would seem like a better test, in my view)?

- There are some missing citations e.g. to the Dreamer line of work (https://arxiv.org/pdf/2301.04104v1.pdf and citations therein), the MuZero line of work (https://arxiv.org/abs/2111.00210 and citations therein), and the aforementioned paper on world modelling in the Harvest game (https://arxiv.org/pdf/2305.11358.pdf).

**Questions:**

See "Weaknesses".

Overall, I think that the motivation for the paper is strong and the Dyna-PSRO algorithm has merit. However, the authors must be more careful to measure the outcomes of social dilemma environments in terms of metrics that make intuitive sense (e.g. collective return) rather than simply measuring deviation from Nash. I hope that they are able to take on board the feedback above to improve the paper, and thus to rigorously demonstrate the benefits of world modelling in PSRO over existing baselines for both zero-sum and general-sum interactions.

---

> ### Author Response · Authors · 2023-11-12
>
> Thank you for taking the time to review our work. We are glad to hear that you found the motivation good, the algorithms well-described, and the results reasonably convincing. We address your specific comments and questions below (paraphrased).
>
> > “Do PSRO or Dyna-PSRO find cooperative solutions in Harvest? If not, why use them?”
>
> Our perspective is that PSRO is a method for finding solutions (e.g., Nash equilibria) for games, and we evaluate our methods based on their computational performance in finding a game solution. In some settings one might prefer to find some solutions (e.g., “cooperative” or high welfare) rather than others; how to address that in PSRO is a topic of much current research but not what we address in this paper. We would argue that being able to find any solution of a complex game is often quite interesting and valuable.
>
> Actually, any equilibrium solution for Harvest will balance cooperative and competitive elements, and we do (anecdotally) observe this in the solutions found by our algorithms. This was qualitatively observed as the agents established a pattern of harvesting the orchard in a circular pattern. Moreover, we found it rare that a policy would take the competitive “tag” action due to it being difficult to effectively learn. We agree that by mentioning in section 3.1 that the “PSRO policies are highly competitive, tending to over-harvest,”  may be misleading. This discussion comment was meant only to provide some intuitive insight about this very small subsample of policy behavior, and not commentary about solutions found by the full game-solving algorithm (PSRO or Dyna-PSRO).
>
> An interesting question to explore in the future is how alternative solution concepts or MSSs affect the advantages and disadvantages of PSRO and Dyna-PSRO.
>
> > “Organization of the paper is confusing.”
>
> We chose to organize the paper in this way because we view section 3, which contains the referred “ablations or analyses”, as a methods section that is essential to understanding any presented results with respect to Dyna-PSRO. The experimental details found within section 3 are directly used by section 4, and any results on Dyna-PSRO would be impossible to interpret without those details.
>
> > “Does not compare to existing baselines.”
>
> The reviewer is correct to note that there exist many works that have studied these games as they appear in MeltingPot. What separates this line of work from them is their goal: the cited work focuses on primarily either studying the game itself or on simultaneously learning policies. Their primary interest isn’t in solving the game for any specific solution (e.g., Nash). To this end, these works likely have extremely strong policies in specific strategic settings, that may perform poorly with other coplayers (i.e., against policies that they were not trained with). This is not because either method is better or worse, but because the goals of these works are different. The PSRO line of work is interested in developing general game-solving algorithms that are notably flexible on solution concepts. Therefore, we think making direction comparisons is unclear for such disparate algorithms, and unfair as they are solving meaningfully different problems.
>
> > In Figure 5, why do the Dyna-PSRO curves terminate before the PSRO curves?
>
> This is because Dyna-PSRO has a longer walltime than PSRO, so we were more limited in how long we were able to run the algorithm. Note that in “Harvest: Categorical” Dyna-PSRO has converged on the x-axis, and that in the other two games, Dyna-PSRO outperforms the other two games when PSRO is allowed to run for many more iterations.
>
> > In Figures 3 and 4, what is the difference between the left-hand plot and the right-hand plot?
>
> The difference between the two plots is in the x-axis. The left-hand plot measures each learning algorithm's performance against  “Real Experiences” used during training; whereas, the right-hand plot measures against both “Real and Planned Experiences.” These terms are introduced in Section 3.2.1 (paragraph one, bolded terms), where real experiences are data from interaction with the real game, and planned experiences are data generated from interaction with the world model. The left-hand plot shows the comparison of learners against the objective of interest since we’re assuming generating real experiences is the bottleneck in game solving. The right-hand plot offers additional insight into the respective algorithms by further comparing them by the amount of additional planned experiences they use. This offers insights into the training dynamics of these learners, and how they tend to take more wall time (as they need many more gradient steps) to run than the baseline methods. We will update the figure captions in this section to make this more clear.

---

> ### Author Response · Authors · 2023-11-12
>
> > “How did the authors arrive at their methodological choices in the strategic diversity section?”
>
>   a) Why are two PSRO policies used? Because each additional policy resulted in combinatorially more results and compute requirements to perform.
>
>   b) What is meant by “the PSRO policies were arbitrarily sampled”? We randomly sampled, without replacement, the policies from the set of policies generated by PSRO.
>
>   c) Why are the PSRO policies subsampled and what is meant by this? From the three policies, per seed, we train our world models on, we take different subsamples of them to train different world models. For example, subsampling only the first policy {1}, or the first and third policy {1, 3}. This was done to show the impact of having more policies contribute to the training data of a world model.
>
>   d) Why is sampling a different policy from PSRO a good test of generalization? It’s unclear whether the specific nature of the learning algorithm that is used to generate the held-out policy is meaningful for this test of generalization. The more pertinent question is how different is the held-out policy from the ones in the train set. In our rebuttal to reviewer c2GE we provide data showing the similarity between the policies, which we believe justifies our choice in using another PSRO policy as a test for generalization.
>
> > “Missing citations”
>
> We will add the suggested citations to the final version of the manuscript, thank you for the suggestions.

---

> > ### Comment · Reviewer_SAe1 · 2023-11-22
> >
> > I thank the authors for their response. Unfortunately I find it fairly unconvincing for the following reasons:
> >
> > 1. "Optimising for Nash". There are very likely a large number of Nash equilibria in this game, treated as a temporally extended interaction. As the folk theorem tells us, subgame perfect Nash equilibria are in fact very plentiful in repeated games. The problem in social dilemmas is not to find an arbitrary Nash equilibrium, but rather to find Nash equilibria which are beneficial for both individuals and the group. So I disagree with the authors that finding a Nash equilibrium is necessarily a useful contribution in a social dilemma.
> >
> > 2. "Comparison to baselines". I disagree with the authors that it is scientifically valid to not compare with baselines. If, as the authors claim, their method is better than baselines at cooperating with novel co-players, then they should demonstrate this empirically. On the other hand, if the authors wish to demonstrate that their world modelling method is useful independently of its benefit for improving the state of the art, then they should show this on real-world data. I don't believe that it is valid to both use artificial domains and to not compare against baselines.
> >
> > 3. "Figures 3 and 4". Many thanks for your explanation of Figures 3 and 4, this now makes sense. However, it raises the question of why the real-environment reward is so low in the simulated experience training phase, and why the model reward drops at the start of the real environment training phase. This would suggest to me that there was some significant distribution shift between the model and the real environment, which is not analysed in detail.
> >
> > 4. "Comparison to https://arxiv.org/pdf/2305.11358.pdf". The authors have not distinguished the ways in which their work has significant impact beyond this prior work to my satisfaction.
> >
> > I hope that the authors are able to take this feedback on board to improve the paper. If there were able to more clearly articulate and empirically validate their research question, then I believe that a future version of the paper could be impactful. However, I cannot recommend acceptance in its current form.

---

> ### Author Response · Authors · 2023-11-22
>
> We thank the reviewer for their reply, and continuing to offer suggestions for improving our manuscript. We address the individual questions and comments below.
>
> > Empirical-Game Learning (EGL) Versus Multiagent Reinforcement Learning (MARL)
>
> We believe the primary point of confusion between us and the reviewer is that they are projecting the problem space of MARL onto our work which is within EGL. In MARL, one is often interested in agents that are simultaneously learning and key problems are how to learn any effective joint policy and what behaviors emerge throughout this process. In EGL, one is interested in modeling an underlying game in a way that affords analytic game reasoning. The primary deliverable is an empirical game that can be solved for as a proxy for the real game. The key problems are how to build the game model such that effectively captures the strategic landscape of the underlying game. The challenges, considerations, and approaches between these two fields is largely disparate. A MARL approach does not produce an artifact that can have game reasoning performed on. Their optimization objectives are different: MARL optimizes for return, and game-solving algorithms optimize for a specified solution concept.  MARL performs only one application of DRL to compute policies, whereas, game-solving algorithms apply _many_ DRL to build a population of policies. This makes their direct comparison unclear.
>
>
> (continues in "Comparison as baslines").
>
>
> > Optimizing for Nash
>
> We agree with the reviewer that there is often many equilibria, and that high-welfare equilibria are particularly interesting in social dilemma games. Finding high-welfare equilibrium is an equally valid problem, but not the one addressed in this work. The primary interest of this work is singularly to reduce the cost, measured in experiences, of performing learning-based game-solving algorithms. Our solution is notably agnostic to the solution concept of interest, and could readily be applied to high-welfare equilibria. This is why we compare against both Harvest and Running With Scissors, as they represent different game classes of interest (one would not be interested in welfare in Running With Scissors). However, finding _any_ equilibria in these more complex games is an active area of research. Moreover, finding equilibria for _specific_ equilibria such as high welfare equilibria is also an activate area of research, and is orthogonal to this study. Never do we claim to be making a contribution to the analysis of social dilemmas, or attempt to solve for the problem of learning high-welfare equilibrium as the reviewer suggests.
>
>
> > Comparison as baselines
>
> We do not think it is fair to characterize our rebuttal as saying that "it is scientifically valid to not compare with baselines". Our rebuttal points out that the suggested baselines are unfit, because they take a meaningfully different problem. Moreover, our work contains a multitude of baselines that are more in line with the problem studied in this paper: how to effectively use planning to reduce the computational cost of game solving. We believe the reviewer is conflating the multiagent reinforcement learning problem of _learning any joint policy_ with the problem studied here of _general game solving_. For examples of similar game-solving algorithm papers see (small sample here):
> - Lanctot, et al.. A Unified Game-Theoretic Approach to Multiagent Reinforcement Learning. NeurIPS'17
> - Balduzzi, et al.. Open-ended Learning in Symmetric Zero-sum Games. ICML'19.
> - McAleer, et al.. Anytime PSRO for Two-Player Zero-Sum Games. arxiv'22.
> - Marris, et al.. Multi-Agent Training beyond Zero-Sum with Correlated Equilibrium Meta-Solvers. ICML'21.
> - McAleer, et al.. Pipeline PSRO: A Scalable Approach for Finding Approximate Nash Equilibria in Large Games. NeurIPS'20.
> - Yao, et al.. Policy Space Diversity for Non-Transitive Games. arXiv'23.
> - Smith, et al.. Strategic Knowledge Transfer. JMLR'23.
> - Liu, et al.. Towards Unifying Behavioral and Response Diversity for Open-ended learning in Zero-sum games. NeuRIPS'21.
> - McAleer, et al.. XDO: A Double Oracle Algorithm for Extensive-Form Games. NeurIPS'21.
> - Perez-Nieves, et al.. Modelling behavioral diversity for learning in open-ended games. ICML'21.
>
> These are a sample of papers that similarly to ours tackle the problem of designing learning-based game-solving algorithms. As is standard, these papers similarly do not compare against multiagent reinforcement learning algorithms, because the problem being solved is different. Instead, they compare against their own self-ablations or other game-solving algorithms which are appropriate baselines, as we argued in our rebuttal. This same practice is adopted in our work.

---

> ### Author Response · Authors · 2023-11-22
>
> > Figures 3 and 4
>
> As we mention in the paper, there is a significant distribution shift between the model and real environment. This is a well known phenomena in world-models that do not operate in latent state space. We include extensive results in the strategic diversity section highlighting the limitations our of world model, which directly contribute to this point (as discussed in the paper). As errors compounding over time is well known, we known and have communicated the reason for this failure. If there is a particular analysis the reviewer thinks would be useful we are happy to try to compile it.
>
>
> > Comparison with workshop paper
>
> The suggested paper is concurrent work. It shares two commonalities with our manuscript: (1) its applies model-based reinforcement learning, and (2) it uses Harvest as an evaluation domain. It is **different** in almost every other aspect:
> - Problem statement:
> 	- Workshop: focused specifically on the emergent behavior of agents in social dilemmas.
> 	- Manuscript: focused on the development of general game-solving algorithms
> - Method (high-level):
> 	- Workshop: applies multiagent reinforcement learning to learn a single joint strategy.
> 	- Manuscript: applies empirical game theory to learn an empirical game and solution.
> - Method (low-level):
> 	- Workshop:
> 		- Computes policies using concurrent learners.
> 		- Uses online reinforcement learning.
> 		- Uses latent world models.
> 	- Manuscript:
> 		- Computes an _empirical game_, using RL as a subroutine with only ever one concurrent learner.
> 		- Uses offline reinforcement learning.
> 		- Uses a non-latent world model.
>
>
> > "more clearly articulate and empirically validate their research question"
>
> The key research question addressed in this paper is to reduce the computational cost of learning-based game-solving algorithms through transferring learned world knowledge. We also consider as a minor point the reciprocal benefit of game-solving on learning world models. The major point of the research is shown in Figure 5, and the minor points in Figure 2-4. This is summarized in the last paragraph of the introduction copied here: "The main points of **novelty** of this paper are as follows: (1) empirically demonstrate that world models benefit from the strategic diversity induced by an empirical game; (2) empirically demonstrate that a world model can be effectively transferred and used in planning with new other-players. The **major contribution** of this work is a new algorithm, Dyna-PSRO, that colearns an empirical game and world model finding a stronger solution at less cost than the baseline, PSRO."

---

### Author Response · Authors · 2023-11-12
**General Rebuttal**

We thank all of the reviewers for taking the time to review our manuscript and provide us feedback on how to continue to improve it. We are happy to hear that reviewers found our approach “quite original” (sm74), “intuitively sensible” (SAe1), and addressing a “valid problem in PSRO” (c2GE). Moreover, we thank the reviewers for their kind words on the content and presentation of the manuscript as “well-described” (SAe1), “well-presented and well-executed” (c2GE),  and “well-written” (qJSG).

We have responded to each reviewer’s individual comments and questions in their respective threads. We are happy to continue to answer any other questions that the reviewers may have during the discussion period.

---

### Meta-Review · Area_Chair_ZAoW · 2023-12-07

**Metareview:**

This paper proposes jointly learning world models and empirical game models. This relatively novel proposal sparked both interest and confusion among reviewers suggesting opportunities to improve the clarity of the paper could help its adoption by and future impact on the research community. The empirical results could also be improved by comparison to more recent baselines and stronger justification for hyper-parameter settings.

**Justification For Why Not Higher Score:**

+ Multiple reviewers had remaining concerns about the strength of the empirical results
+ Only reviewer remaining in support gave limited reasoning for disagreeing the experiments need further work

**Justification For Why Not Lower Score:**

N/A

---

### Decision · Program_Chairs · 2024-01-16

Reject